# SPIN1 facilitates chemoresistance and HR repair by promoting Tip60 binding to H3K9me3

Yukun Wang, Mengyao Li, Yuhan Chen, Yuhan Jiang, Ziyu Zhang, Zhenzhen Yan [ID], Xiuhua Liu [ID] [✉] & Chen Wu [ID] [✉]

## Abstract

**The tandem Tudor-like domain-containing protein Spindlin1 (SPIN1) is a transcriptional coactivator with critical functions in embryonic development and emerging roles in cancer. However, the involvement of SPIN1 in DNA damage repair has remained unclear. Our study shows that SPIN1 is recruited to DNA lesions through its N-terminal disordered region that binds to Poly-ADP-ribose (PAR), and facilitates homologous recombination (HR)-mediated DNA damage repair. SPIN1 promotes H3K9me3 accumulation at DNA damage sites and enhances the interaction between H3K9me3 and Tip60, thereby promoting the activation of ATM and HR repair. We also show that SPIN1 increases chemoresistance. These findings reveal a novel role for SPIN1 in the activation of H3K9me3-dependent DNA repair pathways, and suggest that SPIN1 may contribute to cancer chemoresistance by modulating the efficiency of double-strand break (DSB) repair.**

**Keywords** SPIN1; DNA Damage Repair; H3K9me3; Tip60; ATM
**Subject Categories** Cancer; DNA Replication, Recombination & Repair

## Introduction

During the cell cycle, cells encounter significant internal and external hazards that have the potential to damage DNA (Chatterjee and Walker, 2017; Lindahl and Barnes, 2000). DNA damage response (DDR) is a crucial cellular process that plays a key role in maintaining genome integrity. Defects in DNA repair can lead to genomic instability and contribute to the development of various diseases, including cancer (Lord and Ashworth, 2012). Moreover, DDR is also an important determinant of cell sensitivity to chemotherapeutic drugs, as alterations of the DNA damage-signaling pathway can confer chemoresistance in cancer cells.

SPIN1 is a 262 amino acid protein composed of an N-terminus intrinsically disordered region (IDR) and three tudor-like domains, with each domain containing approximately 50 amino acids (Zhao et al, 2007). The Tudor domain functions as a "reader" module that recognizes histone methylation. Structural studies have revealed that the second Tudor-like domain of SPIN1 can recognize

H3K4me3 or H4K20me3, while the first Tudor module can recognize H3R8me2a or H3K9me3. This indicates that SPIN1 acts as a multifunctional histone reader capable of recognizing H3K4me3, H4K20me3, H3"K4me3-R8me2a" and H3K4me3-K9me3 (Du et al, 2021; Su et al, 2014; Wang et al, 2018; Yang et al, 2012). SPIN1 has been extensively studied for its role as a transcriptional coactivator in embryonic development and its implications in cancer. Research has shown that SPIN1 binds to H3K4me3, promoting its binding in the promoter region of rDNA, and facilitating gene transcription, including rDNA, IL1B, and BST2, etc (Bae et al, 2017a; Wang et al, 2011; Yang et al, 2012). In addition, SPIN1 can bind to H3K4me3R8me and activate protein arginine methyltransferase 2 (PRMT2) and MLL complex, thereby promoting downstream Wnt/β-catenin signaling (Su et al, 2014). The third tudor-like domain of SPIN1 can bind to SPINDOC, disrupting its ability to read chromatin methylation, resulting in its detachment from chromatin, indicating that the SPINDOC-SPIN1 complex can act as a transcriptional repressor (Bae et al, 2017a; Devi et al, 2019). Of note, SPIN1 also binds to H3K4me3K9me3, displacing HP1 proteins from H3K4me3K9me3-enriched rDNA loci and facilitating the transcription of rDNA repeats (Du et al, 2021). Recently, we discovered that phase separation of SPIN1, mediated by its IDR, regulates the recognition of SPIN1 to H3K4me3, suggesting the crucial role of its IDR in SPIN1's functions (Wang et al, 2024).

Furthermore, SPIN1 is highly expressed in various types of cancers (Chen et al, 2018; Franz et al, 2015; Wang et al, 2012), suggesting its potential role as a tumor promoter. Previous studies have demonstrated the oncogenic property of SPIN1, which may be attributed to its negative regulation of uL18, leading to p53 inactivation (Fang et al, 2018). Excessive levels of SPIN1 may have detrimental effects on spindle microtubule organization and chromosomal stability, potentially contributing to cancer development (Wang et al, 2012; Yuan et al, 2008; Zhang et al, 2008). Accumulating evidence indicates that SPIN1 promotes tumorigenesis by activating multiple cancer-related downstream signaling pathways, such as the Wnt, PI3K/AKT, and RET pathways (Chen et al, 2016; Devi et al, 2019; Franz et al, 2015). Given the link between histone H3K9 methylation and DNA damage detection (Sun et al, 2009), it is speculated that SPIN1 may be involved in DNA damage repair. However, the direct evidence regarding the involvement of SPIN1 in DNA damage repair remains largely unexplored.

College of Life Sciences, Hebei University, Baoding, Hebei Province 071002, China. ✉E-mail: liuxiuhua_2004@163.com; wuchen@hbu.edu.cn

ATM kinase functions as a critical protein in sensing and repairing DNA double-stranded breaks (DSBs) (Bakkenist and Kastan, 2004; Caporali et al, 2004; Gatei et al, 2000; Nadkarni et al, 2012). The activation of ATM is regulated by multiple factors, including Tip60 acetyltransferase activity and the status of histone H3 lysine 9 trimethylation (H3K9me3) (Sun et al, 2005; Sun et al, 2009). DNA damage triggers rapid acetylation of ATM through a mechanism dependent on the Tip60, a histone acetyltransferase (HAT), which leads to ATM autophosphorylation on Ser 1981 (Sun et al, 2005; Sun et al, 2009). Tip60 plays a crucial role in linking DNA strand breaks in chromatin to the activation of ATM (Sun et al, 2005). The acetyltransferase activity of Tip60 is attributed to its interaction with histone H3 trimethylated on lysine 9 (H3K9me3) at the damage sites (Han et al, 2018; Sun et al, 2009). A reduced interaction between them leads to impaired ATM activation and widespread defects in DSB repair. Deficiency of H3K9 methylation has been shown to cause ATM phosphorylation deficiency, leading to gene instability and impairing DNA damage repair (Zhang et al, 2016). Considering the previous research on histone H3K9me3 and its impact on ATM activation, as well as the identification of SPIN1 as a potent H3 "K4me3-K9me3" bivalent mark, our study aimed to analyze whether SPIN1 is also involved in the cellular response to DNA damage by modulating the activity of ATM (Du et al, 2021).

In this study, we have revealed a previously unknown function of SPIN1 in coordinating the activation of H3K9me3-dependent DNA repair pathways. We provide evidence that SPIN1 is recruited to DNA lesions through its direct interaction with PAR (Poly-ADP-ribose). SPIN1 binds to H3K9me3, resulting in an enhanced interaction between H3K9me3 and Tip60, thereby facilitating the activation of ATM and homologous recombination (HR)-mediated repair. This interaction among SPIN1, H3K9me3, and Tip60 suggests a potential mechanism by which SPIN1 facilitates the recruitment of DNA repair factors to damaged sites. Importantly, our study also demonstrates that SPIN1 plays a crucial role in conferring resistance to chemotherapy agents. These findings highlight the potential clinical implications of targeting SPIN1 in cancer treatment.

## Results

### SPIN1 was rapidly recruited to DNA damage sites

To investigate the role of SPIN1 in DNA damage repair, we first induced DNA damage using laser micro-irradiation and observed the recruitment of endogenous SPIN1 to the damage sites, where it co-localized with γH2AX (Fig. 1A). To further analyze the recruitment kinetics of SPIN1, we conducted live cell imaging by transfecting U2OS cells with a GFP-tagged SPIN1. Our findings demonstrated that SPIN1 was rapidly recruited to sites of DNA damage within 30 s following laser microirradiation (Fig. 1B). Notably, SPIN1 consists of an N-terminus intrinsically disordered region (IDR) and three tander tudor-like domains. Previous results have indicated that the first and second tudor domains are responsible for recognizing histone methylation, while the third tudor-like domain interacts with SPINDOC (Bae et al, 2017a; Du et al, 2021; Wang et al, 2018). To identify the critical domain mediating SPIN1's rapid recruitment at laser-irradiated sites, we

generated GFP-tagged deletion mutants and analyzed GFP fluorescence intensity (Figs. 1C and EV1). Our analysis indicated that the absence of the first (51–125aa), second domain (125–190aa), or third (190–262aa) domain resulted in a slight impairment of the laser stripe compared to the wild-type. However, deletion of the N-terminus IDR (1–50aa) abolished the recruitment of SPIN1 to DNA damage sites, indicating the importance of IDR for the recruitment process. Collectively, our results indicate that SPIN1 plays a crucial role in DNA damage response, with its recruitment to the damage sites predominantly dependent on the N-terminus IDR.

### SPIN1 is recruited to DNA damage sites through direct binding with PAR

Next, we aimed to investigate the mechanism underlying the recruitment of SPIN1 to DNA damage sites. We found that the relocation kinetics of SPIN1 to DNA damage sites were similar to that of PARylation, an early signal generated at DNA lesions (Fig. 2A) (Gibson and Kraus, 2012). To confirm whether DNA damage-induced PARylation mediates the recruitment of SPIN1, we examined the relocation kinetics of GFP-SPIN1 to DNA damage sites in the presence of a PARP inhibitor. Our results demonstrated that the recruitment of SPIN1 was abolished when PAR synthesis was inhibited by Olaparib treatment (Fig. 2B). In addition, we investigated the recruitment of SPIN1 in PARP1-deficient U2OS cells and observed disrupted recruitment of SPIN1, which further supported our hypothesis (Fig. 2B). These findings collectively suggest that the early recruitment of SPIN1 is mediated by DNA damage-induced PARylation.

Interestingly, PAR, an oligosaccharide, is synthesized by PARPs at DNA lesions within seconds in response to DNA damage. To examine the in vivo interaction between SPIN1 and PAR, we performed immunoprecipitation (IP) and dot blotting assays. Reciprocal immunoprecipitation (IP) confirmed the interaction between SPIN1 and PAR following ionizing radiation (IR) treatment (Fig. 2C–F). In addition, we investigated whether the SPIN1-PAR interaction is direct. To test this, we synthesized and purified poly(ADP-ribose) (PAR) and produced recombinant proteins of SPIN1, including the wild type and a deletion mutant lacking amino acids 1–50. Pull-down assays were conducted, and the results showed that GST-SPIN1, but not GST alone, specifically bound to PAR (Fig. 2G), indicating the enhanced pull-down efficiency associated with the presence of IDR. Furthermore, the SPIN1 mutant lacking residues 1–50aa failed to bind to PAR, emphasizing the crucial role of the N-terminus of SPIN1 in its interaction with PAR. Notably, Olaparib disrupted the interaction between SPIN1 and PAR following IR treatment (Fig. 2H), providing further evidence supporting the specific interaction between SPIN1 and PAR, as well as the essential role of the IDR in SPIN1's involvement in the DNA damage response. Taken together, these findings suggest that SPIN1 directly binds to PAR and is required for the early recruitment of DNA lesions.

### SPIN1 promotes HR-mediated repair pathway and enhances the activation of ATM

We next investigated the role of SPIN1 in DSB repair pathways, including homologous recombination (HR) and non-homologous

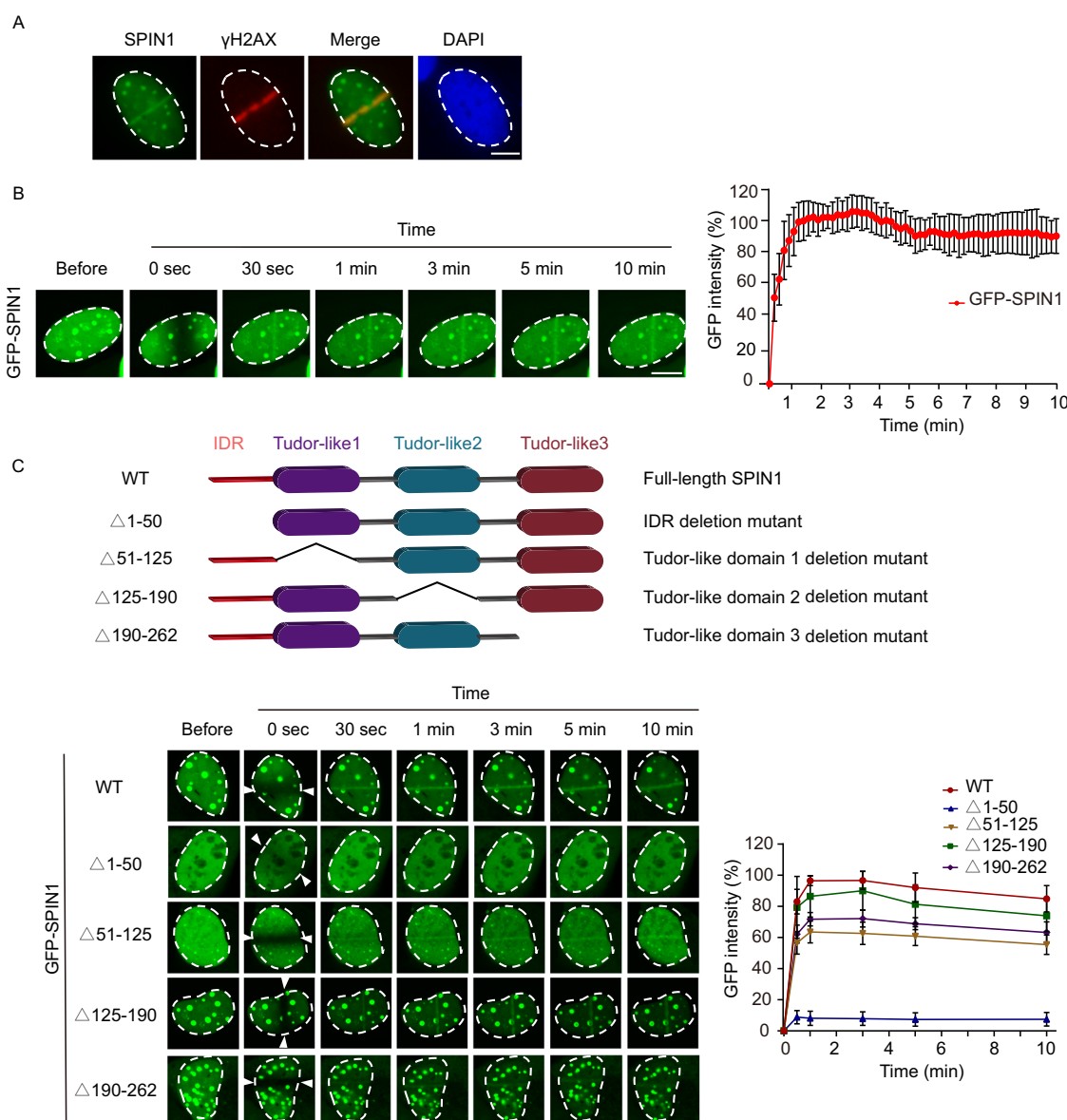

**Figure 1. Rapid recruitment of SPIN1 to DNA lesions.**

(**A**) Localization of SPIN1 at DNA damage sites. DNA lesions were induced by laser microirradiation in U2OS cells and the co-localization between the endogenous SPIN1 and γH2AX were analyzed using immunostaining. Scale bar = 2.5 μm. (**B**) Kinetics of GFP-SPIN1 relocation to DNA damage sites. U2OS cells expressing GFP-tagged SPIN1 were subjected to laser microirradiation, and the relocation kinetics were monitored in a time course. Live cell imaging was conducted at 10-s intervals for up to 10 min after laser-induced DNA damage. Kinetic analysis was performed using CellSens software (Olympus). The data represents the mean ± SD. Scale bar = 5 μm. *n* = 3 biological replicates. (**C**) Relocation kinetics of GFP-tagged SPIN1 and its mutants to DNA damage sites. Stripe formation by SPIN1-WT and deletion mutants (Δ1–50, Δ51–125, Δ125–190, Δ190–262) were observed at different time points following laser microirradiation in U2OS cells. The white arrows indicate the location of DNA damage site. The GFP signal intensities at the laser lines were quantified using Image J software. The data represents the mean ± SD from three independent biological replicates. Scale bar = 5 μm. Source data are available online for this figure.

end joining (NHEJ). Using GFP reporter assays, we found that knockdown of SPIN1 reduced the GFP signaling associated with HR repair pathway, while having no significant effect on NHEJ repair pathway (Fig. 3A,B). In addition, we examined the potential impact of SPIN1 on cell cycle progression. Knockdown of SPIN1 resulted in a slight reduction in the S-phase and increase in G1 phase in HeLa and HEK293T cells by flow cytometry analysis (Fig. EV2A), which are consistent with previous studies

(Yuan et al, 2008; Zhang et al, 2008; Zhao et al, 2007; Lv et al, 2020). However, the slight alteration in the cell cycle cannot fully account for the significant defect in HR observed in SPIN1-depleted cells, further indicating the direct impact of SPIN1 on DSBs repair. To further determine the role of SPIN1 in HR and NHEJ repair processes, we treated U2OS cells with IR and conducted immunofluorescence staining to quantify the foci of key factors involved in HR repair pathway (RAD51 and BRCA1) and

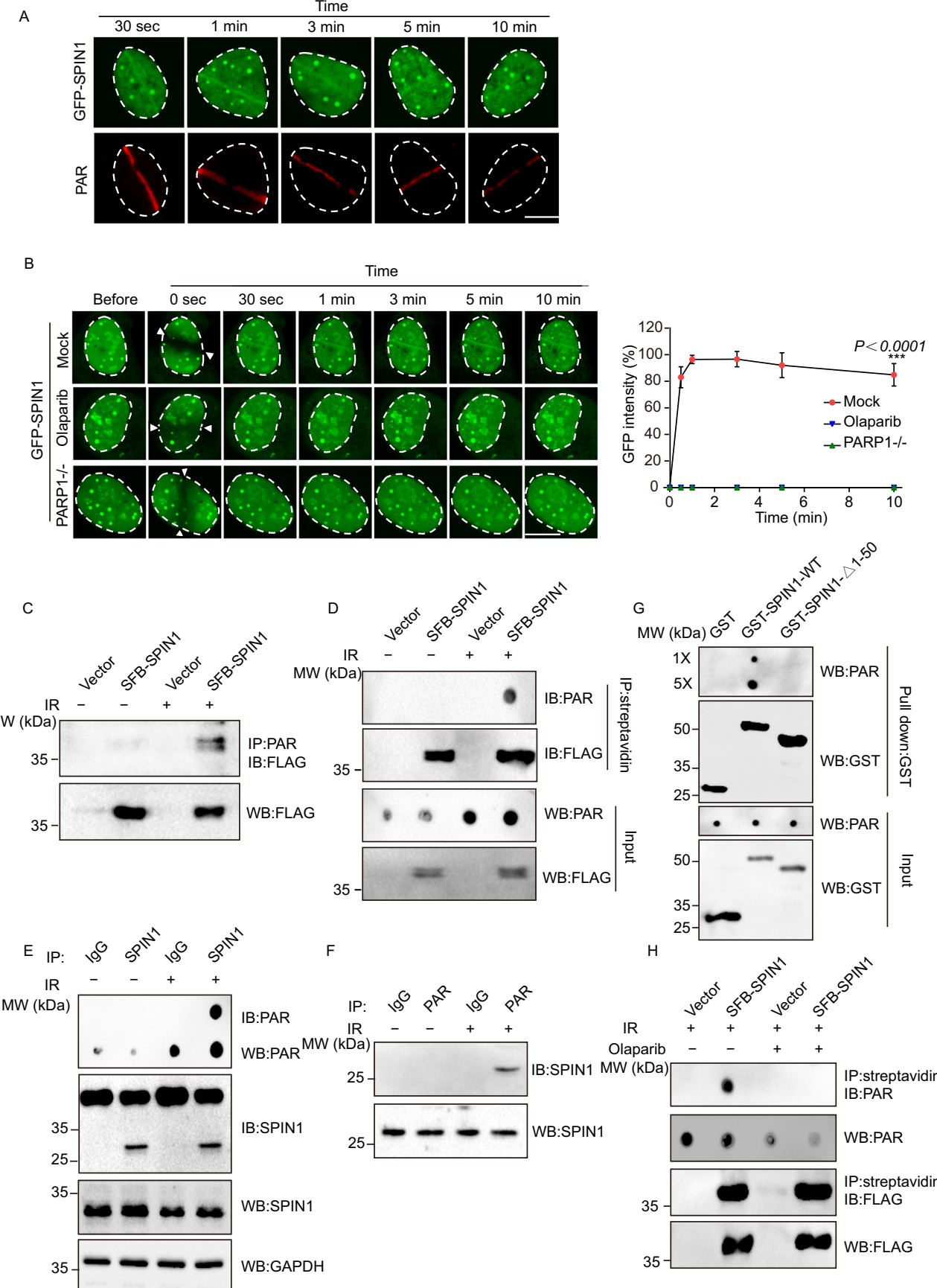

◄   **Figure 2.   SPIN1 directly binds to PAR and is recruited to DNA lesions by PAR.**

(**A**) Representative images show the co-localization of GFP-SPIN1 with PAR at different time points. U2OS cells were transfected with GFP-SPIN1, and 24 h later, DNA damage was induced by laser microirradiation. After the indicated recovery time, the cells were fixed, and immunofluorescent staining was performed using an antibody against PAR. Scale bar =10 μm. (**B**) The effect of Olaparib treatment or depletion of PARP1 on the recruitment kinetics of SPIN1 to DNA damage sites. GFP-SPIN1 was expressed in PARP1-deficient U2OS (PARP1⁻/⁻) or U2OS cells treated with the PARP1 inhibitor (Olaparib). The relocation of GFP-SPIN1 was monitored in a time course following laser microirradiation. White arrows indicate the laser stripe induced by microirradiation. Statistical significance was determined using one-way ANOVA followed by the Tukey Kramer test. Data represent the mean ± SD from three independent biological replicates. ***P < 0.001. Scale bar = 10 μm. (**C, D**) Interaction of SPIN1 with PAR in IR-induced DNA damage. Cell extracts from HEK293T-expressing SFB-SPIN1 or SFB-Vector were immunoprecipitated with an anti-PAR antibody (**C**) or streptavidin beads (**D**), and then analyzed by immunoblotting. (**E, F**) In vivo interaction between SPIN1 and PAR was examined by co-immunoprecipitation (Co-IP) and reciprocal co-immunoprecipitation (Co-IP). HEK293T cells were either treated with or without 10 Gy of IR, and 10 min after IR treatment, the cells were lysed and analyzed using indicated antibodies. Samples of input or immunoprecipitated proteins were analyzed by Western blotting or dot blotting. (**G**) In vitro interaction between SPIN1 and PAR was examined by Pull-down assay. Indicated recombinant proteins were purified from *Escherichia coli* BL21 (DE3) cells. Recombinant GST-fusion proteins were incubated with PAR and subsequently pulled down using glutathione agarose beads. Pull-down samples with normal concentration (1×) or a 5× concentration were analyzed by dot blotting. GST was used as negative control. (**H**) The interaction between SFB-SPIN1 and PAR was examined by co-immunoprecipitation in the presence or absence of Olaparib (100 nM) and IR treatment (10 Gy) with the indicated antibodies. Source data are available online for this figure.

NHEJ repair pathway (53BP1). Our findings demonstrated that the knockdown of SPIN1 resulted in a reduction in the number of RAD51 and BRCA1 foci, while the number of 53BP1 foci remained unchanged, indicating the specific impact of SPIN1 on the HR repair pathway. We found no significant difference in the formation of γH2AX foci between siNC and siSPIN1 groups (Fig. 3C–H). In addition, the immunofluorescence images of untreated cells demonstrated that in the absence of IR treatment, the siRNA treatments did not induce any damage (Fig. EV2B).

The ataxia-telangiectasia mutated (ATM) and ataxia-telangiectasia mutated- and Rad3-related (ATR) protein kinases act as master regulators of the DNA damage response. They regulate their kinase activities to orchestrate a large network of cellular processes to maintain genomic integrity (Maréchal and Zou, 2013). First, we investigated the impact of SPIN1 on the activation of ATR and ATM using Western blotting assay. Our results demonstrated that knockdown of SPIN1 resulted in decreased levels of phosphorylated ATM (P-ATM), while no significant changes were observed in phosphorylated ATR (P-ATR) levels following DNA damage (Fig. EV2C). These findings indicate that SPIN1 primarily promotes the activation of ATM rather than ATR. Consequently, our study focused on elucidating the mechanism through which SPIN1 activates ATM. Next, to further investigate the involvement of SPIN1 in this process, we quantified and compared the number of phosphorylated ATM (P-ATM) foci in SPIN1 knockdown and control cells treated with IR using immuno-fluorescent staining. The results showed a significant decrease in the number of P-ATM foci upon SPIN1 knockdown (Fig. 3I,J). Furthermore, Western blot experiments demonstrated that SPIN1 knockdown led to reduced levels of phosphorylated ATM (P-ATM) and phosphorylated CHK2 (P-CHK2) after DNA damage, indicating a decreased phosphorylation activity of ATM (Fig. 3K,L). Our previous studies have shown that the N-terminal region of SPIN1 is responsible for PAR binding and recruiting SPIN1 to sites of DNA damage. Therefore, we sought to investigate the phosphorylation activity of ATM in cells overexpressing wild-type SPIN1 or a mutant lacking the N-terminal residues (△1–50). Remarkably, we observed that the overexpression of SPIN1 led to increased levels of phosphorylated ATM (P-ATM) and phosphorylated CHK2 (P-CHK2) as compared to cells transfected with the empty vector or the △1–50 mutant (Fig. 3M,N). Collectively, our findings indicate that SPIN1 promotes the activation of ATM, with the N-terminal IDR of SPIN1 playing a role in regulating this activation process.

## SPIN1 facilitates the association of H3K9me3 with Tip60

Previous studies have reported SPIN1 functions as a potent H3 "K4me3-K9me3" bivalent mark reader, thereby balancing gene expression and silencing in H3K9me3-enriched regions (Du et al, 2021). Using Co-IP assays, we observed a significant binding of SPIN1 to H3K9me3 following IR treatment (Fig. 4A). It is known that oncometabolites, such as 2-hydroxyglutarate (2HG), succinate, and fumarate, inhibited the activity of the lysine demethylase KDM4B, leading to aberrant hypermethylation of H3K9 at loci surrounding DNA breaks. This impairs the recruitment of Tip60 and ATM at the DNA Damage sites (Sulkowski et al, 2020). We aimed to investigate whether the binding of SPIN1 to H3K9me3 regulates the function of H3K9me3 in coordinating activation of Tip60-dependent DNA repair pathways. To explore this, we utilized laser micro-irradiation to measure the dynamics of H3K9me3 at DNA damage sites induced in a stripe across the nucleus. Consistent with previous studies (Sulkowski et al, 2020), we observed that H3K9me3 was locally relocalized at the stripe and co-localized with γH2AX at 3 min after laser micro-irradiation. However, SPIN1 knockdown cells showed a decreased accumulation of H3K9me3 at 30 min after laser micro-irradiation, indicating the crucial role of SPIN1 in preserving the presence of H3K9me3 at the site of damage (Fig. 4B,C). Furthermore, in the control group, there was a sustained increase in H3K9me3 levels within three hours after IR treatment, whereas H3K9me3 levels gradually decreased in the SPIN1-knockdown group (Fig. 4D). Notably, H3K9me3 levels were already significantly impaired at one hour post-damage in the SPIN1-knockdown cells, suggesting that the binding of SPIN1 to H3K9me3 protects its methylated modification (Fig. 4D).

It is known that Tip60 directly interacts with H3K9me3, activating its acetyltransferase activity, and promoting the activation of ATM (Han et al, 2018; Sun et al, 2009). To investigate whether SPIN1 regulates the interaction between Tip60 and H3K9me3, we transfected cells with siNC or siSPIN1 and performed a Co-IP assay to examine the interaction between Tip60 and H3K9me3. Our results demonstrated that knockdown of SPIN1 led to a decrease in the binding of endogenous Tip60 to H3K9me3 during the DNA damage (Fig. 4E). Moreover, compared to control cells, we observed an increased interaction between Tip60 and H3K9me3 in cells overexpressing SPIN1 (Fig. 4F), indicating that SPIN1 enhances the interaction between H3K9me3 and Tip60. In addition, we found that knockdown of SPIN1

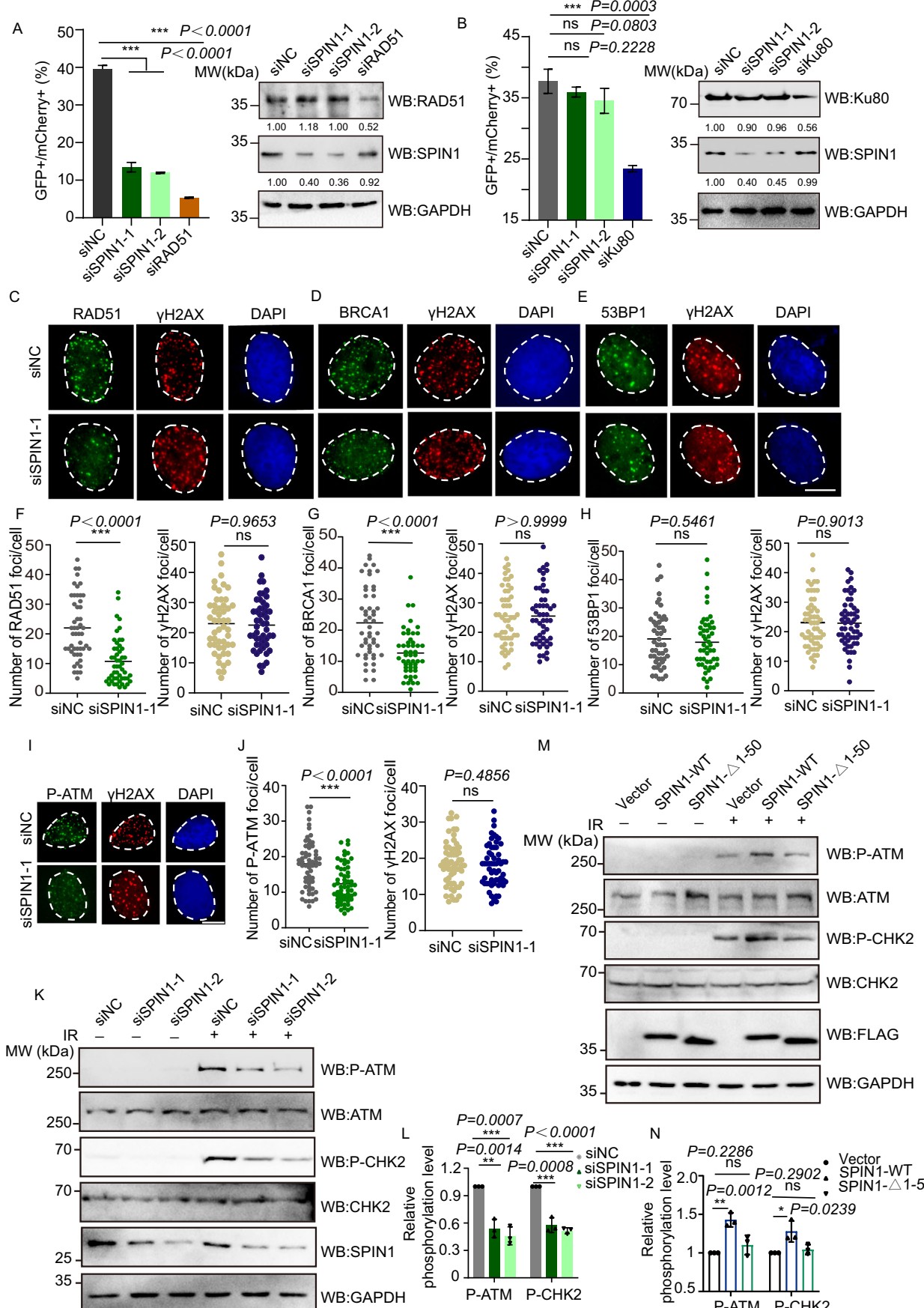

Figure 3. SPIN1 promotes the activation of ATM and HR repair.

(A, B) Knockdown of SPIN1 significantly decreased HR repair. The GFP reporter systems were utilized to examine NHEJ and HR repair mechanisms. The repair efficiency for each mechanism was determined by calculating the percentage of GFP+/mCherry+ cells using flow cytometry (left). Cells were treated with indicated siRNA, and analyzed by immunoblotting (right). The RAD51 and Ku80 siRNAs were used as positive controls for HR and NHEJ, respectively. The significance of differences was evaluated using Student's t-test. Data represent the mean ± SD from three independent biological replicates. (C–E) Immunofluorescence analysis of the foci formation of RAD51, BRCA1, and 53BP1 after IR-induced DNA damage. U2OS cells were transfected with the indicated siRNA, then treated with 10 Gy IR to induce foci formation, which was detected by Immunofluorescence. The representative images of RAD51 (C), BRCA1 (D), and 53BP1 (E) foci were shown. Scale bar = 5 μm. (F–H) The number of foci was counted from three independent experiments with ≥50 randomly selected cells. The data represent the mean ± SD. Statistical significance was determined by the Student's t-test. (I, J) Knockdown of SPIN1 impaired the formation of P-ATM foci. U2OS cells were transfected with indicated siRNA. Then, the cells were treated with 10 Gy of IR, and the formation of P-ATM foci was detected by Immunofluorescence (I). The number of foci was counted from three independent experiments with ≥50 randomly selected cells (J). Statistical significance was determined using the Student's t-test. Data represent the mean ± SD, scale bar = 5 μm. (K, L) The phosphorylation levels of ATM and CHK2 were lower in SPIN1-knocked down cells following IR treatment. HEK293T cells were transfected with the indicated siRNA, treated or untreated with 10 Gy IR and then subjected to Western blot for detecting the indicated proteins (K). Quantitative statistical analysis was performed on phosphorylation levels of ATM and CHK2 from three independent experiments (L). Statistical significance was determined using the Student's t-test. Data are presented as mean ± SD, $n = 3$ biological replicates. (M, N) Overexpression of SPIN1-WT, but not the 1–50 amino acid deletion mutant, promoted the activation of ATM upon DNA damage. HEK293T cells expressing the vector, SFB-SPIN1-WT or SFB-SPIN1-△1–50 were treated or untreated with 10 Gy IR. Total cell lysates were harvested and subjected to immunoblotting with the indicated antibodies (M). Quantitative statistical analysis was performed on phosphorylation levels of ATM and CHK2 from three independent experiments (N). Statistical significance was determined using the Student's t-test. Data are presented as mean ± SD, $n = 3$ biological replicates. *$P < 0.05$; **$P < 0.01$; ***$P < 0.001$; ns, not significant. Source data are available online for this figure.

reduced the acetylation of ATM (Fig. 4G,H). Considering that the first Tudor domain of SPIN1 is responsible for binding with H3K9me3, we further examined whether a deletion mutant of this domain impaired ATM activation. As expected, this mutant weakened the phosphorylation activation of ATM, further confirming the significance of SPIN1's binding to H3K9me3 in the damage response (Fig. EV3A,B). Taken together, these findings suggest that SPIN1 promotes the interaction between Tip60 and H3K9me3, facilitating the activation of ATM and highlighting its significance in DNA damage response.

## SPIN1 enhances DNA damage repair and chemoresistance

Our study has demonstrated the involvement of SPIN1 in HR-mediated DSB repair. To further investigate the biological functions of SPIN1 in the context of DSB repair, we examined the repair kinetics of DSBs using comet assays under neutral conditions after ionizing radiation treatment. Our results showed that loss of SPIN1 significantly impaired DSB repair, while the reintroduction of wild-type SPIN1, but not the IDR deletion (△1–50) or first Tudor domain deletion mutant (△51–125) in SPIN1 knockdown cells, rescued the deficiency in DSB repair (Figs. 5A and EV4A). In addition, we performed colony formation assays to assess cell viability after exposure to different doses of IR in cells treated with either control or siRNA-SPIN1. Knockdown of SPIN1 resulted in fewer cell colonies compared to both the control group and the SPIN1 rescue group. Notably, the restoration of these mutants failed to rescue the deficiency in cell growth, indicating an increased sensitivity of cells lacking SPIN1 to DNA damage (Fig. 5B). Taken together, these findings highlight the significant role of SPIN1 in DNA damage repair and cell survival following DNA damage.

It has been extensively reported that SPIN1 is highly expressed in various types of cancers (Chen et al, 2018; Franz et al, 2015; Wang et al, 2012), suggesting its potential role as a tumor promoter. To further investigate this, we utilized the GEPIA (Gene Expression Profiling Interactive Analysis) web server (http://gepia.cancer-pku.cn/) and observed significantly elevated expression levels of SPIN1 across various cancer types compared to

normal tissues (Fig. EV4B). Previous studies have shown that targeting SPIN1 can enhance the radiosensitivity of non-small cell lung cancer (NSCLC) cells and improve the efficacy of radiotherapy (Jin et al, 2021). Considering the significance of DNA damage response in chemoresistance, we investigated whether SPIN1 may contribute to cancer chemoresistance by affecting the efficiency of DSB repair. To assess this, we treated HeLa and SGC7901 cells with Cisplatin and Olaparib, respectively, and evaluated cell survival. Our results showed that knockdown of SPIN1 increased the sensitivity of cancer cells to Cisplatin and Olaparib, while reintroducing SPIN1 restored their resistance to these drugs (Fig. 5C,D). Surprisingly, we observed that SPIN1 was not recruited to DNA damage sites following Olaparib treatment (Fig. 2B), yet its reintroduction still restored resistance to Olaparib. This suggests that there might be alternative mechanisms through which SPIN1 contributes to cell survival in the presence of Olaparib. Further investigations are needed to determine the specific mechanisms involved. In addition, we investigated SPIN1-mediated chemoresistance using a xenograft mouse model. Consistent with our cell viability assay, the depletion of SPIN1 resulted in a significant increase in sensitivity to Cisplatin and Olaparib, as demonstrated by the remarkable decrease in tumor growth rate, weight, and size (Figs. 5E–J and EV4D, E). Taken together, these results indicate that the depletion of SPIN1 has the potential to enhance tumor chemosensitivity. Our findings suggest that targeting SPIN1 could be a potential strategy to enhance the sensitivity of cancer cells to DNA damage-inducing agents, including IR and anticancer drugs.

## Discussion

SPIN1 has been extensively studied for its role as a histone methylation reader and a transcriptional coactivator that facilitates the expression of rRNA genes, embryonic development and its implications in cancer (Wang et al, 2011; Yang et al, 2012). However, the involvement of SPIN1 in DNA damage repair has remained largely unexplored. Our study demonstrates that SPIN1 is recruited to DNA lesions by directly binding with PAR, thereby promoting HR-mediated DNA damage repair. Furthermore, we discovered that SPIN1 plays a crucial role in maintaining the

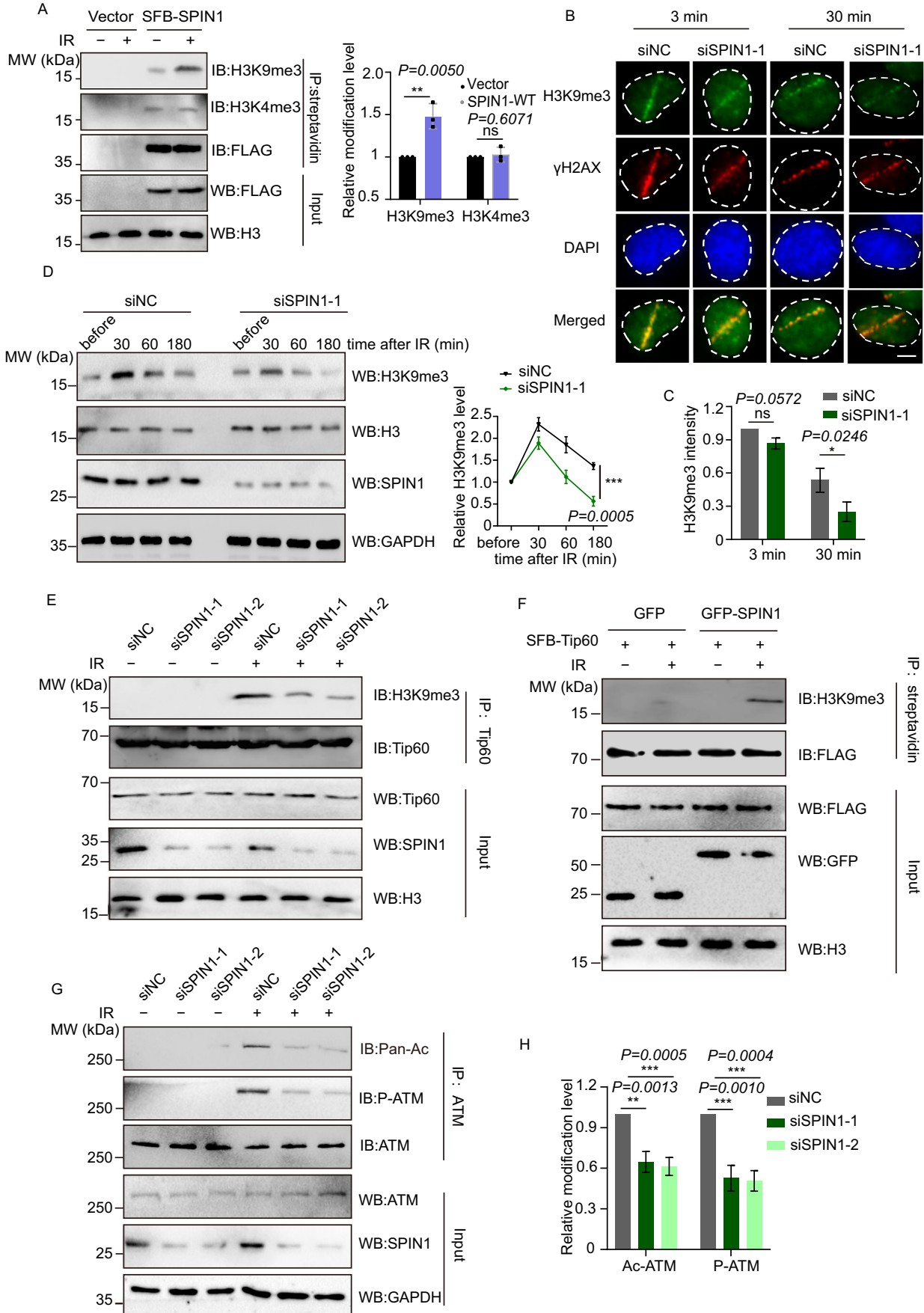

**Figure 4. SPIN1 enhances the interaction between H3K9me3 and Tip60.**

(A) The interaction between SFB-SPIN1 and H3K9me3 were promoted by DNA damage. HEK293T cells expressing SPIN1 or control vector were subjected to co-immunoprecipitation using streptavidin beads and western blot analysis was performed using the indicated antibodies (left). The data are summarized from three independent biological replicates (right). Statistical significance was determined using the Student's t-test. Data are presented as mean ± SD, $n = 3$ biological replicates. (B, C) Representative images showed the co-localization between the H3K9me3 and γH2AX. U2OS cells were transfected with the indicated siRNA. After 72 h, DNA damage was induced by laser microirradiation. Cells were allowed to recover for the specific time and then fixed for immunofluorescent staining by using antibodies against H3K9me3 and γH2AX (B). Scale bar = 5 μm. Quantitative statistical analysis was performed on H3K9me3 intensity from three independent biological replicates (C). Data are presented as mean ± SD. Statistical significance was determined using the Student's t-test. (D) HEK293T cells were transfected with the siNC or siSPIN1, and harvested at the indicated time points after treatment with 10 Gy of IR. The levels of H3K9me3 were detected by Western blotting (left), and quantitative statistical analysis was performed on H3K9me3 grayscale analysis from three independent biological replicates (right). Data are presented as mean ± SD. Statistical significance was determined using Student's t-test. (E) Knockdown of SPIN1 reduces the binding between H3K9me3 and Tip60 after 10 Gy IR treatment. HEK293T cells were transfected with siNC or siSPIN1, and the cell lysates were immunoprecipitated with anti-Tip60 antibody. Western blot analysis was performed to detect the indicated proteins. (F) Overexpression of SPIN1 promotes the binding of H3K9me3 to Tip60. HEK293T cells were transfected with SFB-Tip60 and GFP-SPIN1 or GFP, and the cell lysates were immunoprecipitated with streptavidin beads. Western blot analysis was performed to detect the indicated proteins. (G, H) Knockdown of SPIN1 reduces the acetylation of ATM. HEK293T cells were transfected with the indicated siRNA, and the cell lysates were immunoprecipitated with anti-ATM antibody. Western blot analysis was performed to detect the indicated proteins (G). The data are summarized from three independent biological replicates (H). Statistical significance was determined using the Student's t-test. Data are presented as mean ± SD. *$P < 0.05$; **$P < 0.01$; ***$P < 0.001$; ns, not significant. Source data are available online for this figure.

stability of H3K9me3, a histone modification associated with DNA repair processes. The binding of SPIN1 to H3K9me3 enhances its interaction with Tip60 and promotes the activation of ATM. This SPIN1-H3K9me3-Tip60-ATM recognition and signaling axis suggests a potential mechanism by which SPIN1 facilitates the recruitment of DNA repair factors to damaged sites. Importantly, our results also demonstrate that SPIN1 confers chemoresistance to DNA-damaging drugs in cancer cells. Therefore, the dysregulation of SPIN1 in cancer cells may disrupt the efficiency of DSB repair, leading to increased resistance to DNA-damaging drugs.

In our study, we discovered that SPIN1 is quickly recruited to DNA lesions by PAR binding, suggesting that SPIN1 participates in PAR-mediated early DNA damage response. Various PAR-binding motifs have been identified in these DNA damage response factors, including PBZ, MACRO, BRCT, FHA, RRM, OB-fold, and PIN domains (Ahel et al, 2008; Gibson and Kraus, 2012; Li and Yu, 2013; Zhang et al, 2014). However, the recruitment of SPIN1 is mediated by its N-terminal IDR. Each ADP-ribose molecule in the PAR chain contains two phosphate moieties that contribute a significant amount of negative charge. This negative charge may facilitate the relaxation of chromatin at DNA lesions by repelling adjacent negatively charged DNA molecules (Caron et al, 2019). Given the enrichment of a cluster of Lys/Arg with positive charge at the N-terminal IDR of SPIN1 (Zhang et al, 2018), we speculate that electrostatic interaction may play an important role in the binding between SPIN1 and PAR. Nevertheless, further research is required to gain a more comprehensive understanding of the binding kinetics between SPIN1 and PAR, particularly through surface plasmon resonance (SPR) or isothermal titration calorimetry (ITC) experiments. Interestingly, we observed a slower recruitment signal of SPIN1 to the damage site when any of three tudor-like domains was deleted. This suggests that, besides the N-terminal IDR of SPIN1, tudor-like domains may also play roles in limiting the DNA damage response.

Elevated levels of SPIN1 protein have been observed in various types of cancer, such as prostate cancer, pancreas cancer, breast cancer, colon cancer and ovarian cancer, etc (Chen et al, 2016; Chew et al, 2013; Fang et al, 2018; Lv et al, 2020; Zhou et al, 2021). Overexpression of SPIN1 has been shown to cause alterations in cell cycle distribution during mitosis and result in chromosomal instability (Li et al, 2021; Wang et al, 2012; Yuan et al, 2008). Given

that SPIN1 promotes transcription of rRNA genes (Wang et al, 2011), it is reasonable to assume that its overexpression in many human cancers facilitates ribosome biogenesis to support the rapid growth and proliferation of cancer cells (Du et al, 2021). In our study, we demonstrated that SPIN1 plays a role in DNA damage repair, which is crucial for maintaining chromosomal stability. The overexpression of SPIN1 in cancer cells enhanced its binding to H3K9me3 and persistently promoted the activation of ATM-mediated DNA repair in cancer cells exposed to DNA-damaging drugs. This resulted in increased cell proliferation and chemore-sistance. The findings of our study provide important insights into not only the role of SPIN1 in DNA damage repair but also its potential implications in cancer chemoresistance. Our study suggests a new therapeutic strategy for blocking SPIN1's binding to H3K9me3 to inhibit tumor growth and chemoresistance.

Previous studies have demonstrated that SPIN1 acts as a reader for histone methylation, specifically binding to various histone codes, including H3K4me3, H3R8me2, H3K9me3 and H4K20me3 (Du et al, 2021; Su et al, 2014; Wang et al, 2018; Yang et al, 2012). H3K9me3 is a well-known marker of heterochromatin and is highly enriched in these regions (Grewal and Jia, 2007). This modification plays a critical role in maintaining the silent and compact conformation of heterochromatin by recruiting HP1, Kap-1, and H3K9 methyltransferases (Grewal and Jia, 2007; Iyengar and Farnham, 2011). However, H3K9me3 can be found in non-heterochromatin regions (Vakoc et al, 2006). Accumulating evidence suggests a direct connection between H3K9me3 status at DSBs and the repair process, as well as the remodeling of the damaged chromatin template. DSBs promote H3K9 methylation, leading to H3K9me3 on large chromatin domains adjacent to the DSBs. This process transiently forms repressive chromatin, stabilizing the chromatin structure and facilitating the activation of DSB-signaling proteins. Tip60 activation at DSBs is dependent on local H3K9 trimethylation, which in turn activates ATM (Ayrapetov et al, 2014; Sun et al, 2009; Williamson et al, 2012). Recently, Sulkowski et al (2020) reported that DSBs induce a rapid increase in H3K9 trimethylation, which coordinates the recruitment of Tip60 and MRE11, promoting ATM activation, licensing end resection, and subsequent recruitment of RPA, BRCA1, and RAD51. Inhibition of the lysine demethylase KDM4B (responsible for demethylating H3K9) by oncometabolite

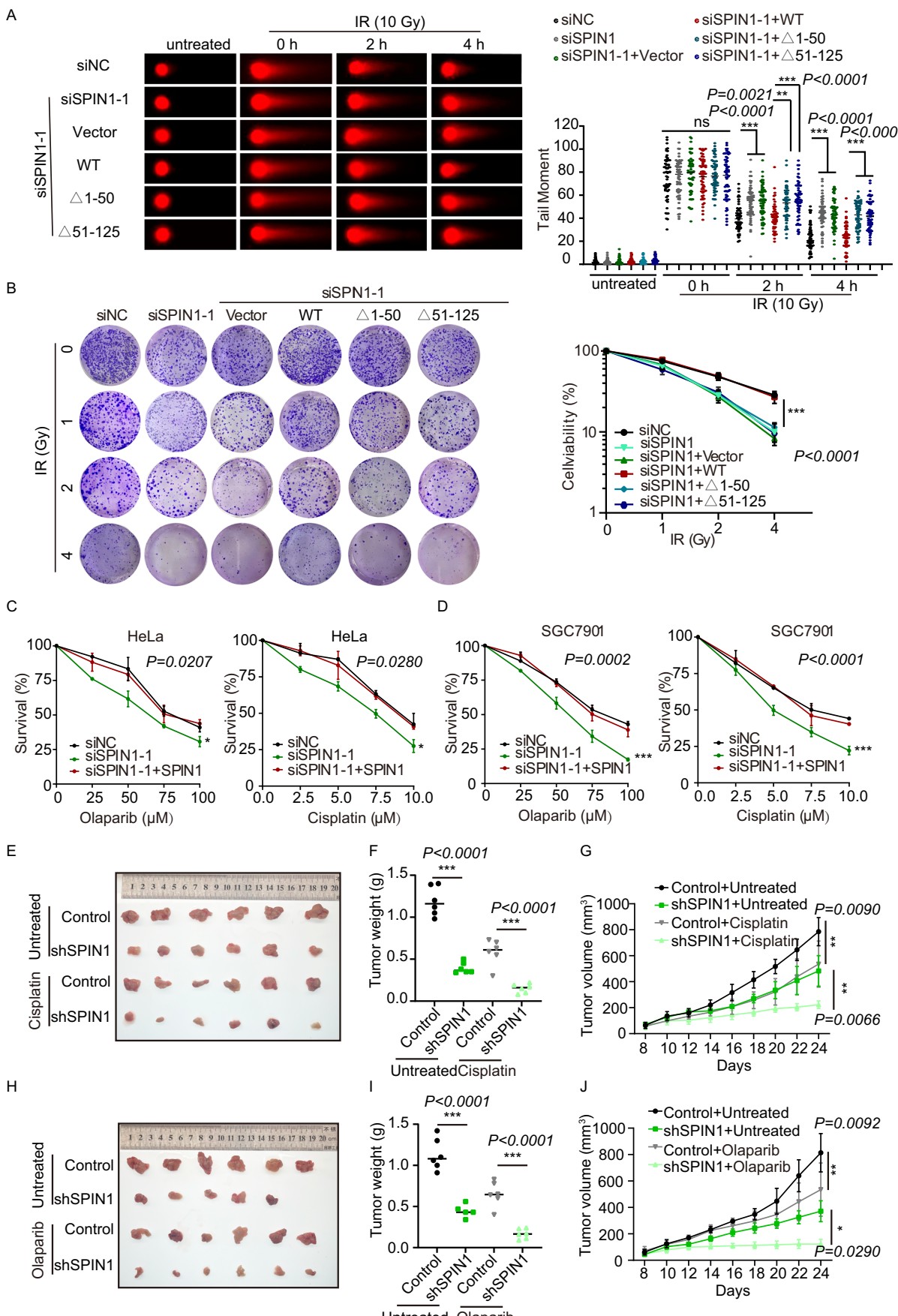

**Figure 5. SPIN1 enhances DNA damage repair and chemoresistance.**

(A) Knockdown of SPIN1 leads to impair DNA damage repair. The neutral comet assay was used to assess the DNA damage induced by 10 Gy of IR in the indicated HEK293T cells. Representative comet images and quantitative analysis of tail moments are presented. Tail moments were summarized from three independent experiments with ≥50 cells. Statistical significance was determined using the one-way ANOVA followed by the Tukey Kramer test. The data are presented as mean ± SD. (B) Knockdown of SPIN1 renders cells more sensitive to IR treatment. HEK293T cells were transfected with the indicated siRNA and plasmids, and colony-formation assays were performed after exposure to the indicated IR dose. Representative images of colonies in plates stained with Giemsa were shown. Statistical significance was determined using one-way ANOVA followed by the Tukey Kramer test. The data represent the mean ± s.e.m. from three independent biological replicates. (C, D) Viability of HeLa (C) or SGC7901 cells (D) was analyzed using the CCK8 assay. HeLa and SGC7901 cells were transfected with the indicated siRNA and plasmids. After 48 h, the cells were seeded in 96-well plates and treated with different concentrations of Cisplatin or Olaparib. Data represent the mean ± SD, $n = 3$ biological replicates. Statistical significance was determined using one-way ANOVA followed by the Tukey Kramer test. (E) The combination of Cisplatin and SPIN1-knockdown significantly suppressed xenograft tumor growth. The image showed the xenograft tumors dissected from Control, shSPIN1, Control+Cisplatin, shSPIN1+Cisplatin groups. (F, G) The tumor weight was measured at the experimental endpoint (F). Tumor volumes were monitored using calipers at the indicated time points (G). $n = 6$/group. Statistical significance was determined using one-way ANOVA followed by the Tukey Kramer test. (H) The combination of Olaparib and SPIN1-knockdown significantly suppressed xenograft tumor growth. The image showed the xenograft tumors in Control, shSPIN1, Control+Olaparib, shSPIN1+Olaparib groups. (I, J) The tumor weight was measured at the experimental endpoint (I). Tumor volumes were monitored using calipers at the indicated time points (J). $n = 6$/group. Statistical significance was determined using one-way ANOVA followed by the Tukey Kramer test. *$P < 0.05$; **$P < 0.01$; ***$P < 0.001$; ns, not significant. Source data are available online for this figure.

(2HG, succinate, and fumarate) leads to aberrant hypermethylation of H3K9 at loci surrounding DNA breaks, masking the essential local H3K9me3 signal required for proper HR-mediated repair (Sulkowski et al, 2020). Therefore, H3K9me3 likely plays a crucial role in linking the chromatin structure at DSBs to the activation of DSB signaling proteins. Furthermore, it is known that SPINDOC interacts with SPIN1 (Bae et al, 2017a; Du et al, 2021; Wang et al, 2018). Previous studies have determined the crystal structures of SPIN1 (aa:50–262) bound to SPINDOC (PDB code: 7CNA for the SPIN1-SPINDOC (aa:253–295) complex; PDB code: 7E9M for the SPIN1-SPINDOC (aa:256–281) complex). However, these studies reported contradictory results (Du et al, 2021; Zhao et al, 2024). Notably, a recent study reported that SPINDOC can bind PARP1 to facilitate PARylation and the DNA damage response independently of SPIN1 (Yang et al, 2021). Consequently, further investigation is required to determine whether SPINDOC directly mediates SPIN1's involvement in DNA damage repair and the precise regulatory mechanism.

Our data demonstrate that SPIN1 plays a crucial role in enhancing the interaction between Tip60 and H3K9me3, thereby effectively promoting the HR-mediated repair pathway. As a result, we propose a working model for our study, suggesting that SPIN1 actively contributes to the maintenance of localized H3K9me3 signaling at DNA double-strand break (DSB) sites. This crucial maintenance mechanism likely involves the prevention of rapid demethylation of H3K9me3 by demethylases, such as KDM4B, as elucidated in the aforementioned study conducted by Sulkowski et al (Fig. 6). In general, histone methylation patterns are frequently altered in human tumors (Bernstein et al, 2007; Seligson et al, 2005). These aberrant histone methylation signatures likely play a crucial role in the development of cancer by regulating the efficiency and extent of DNA repair at specific chromatin sites. Identifying SPIN1 as a key player in H3K9me3-dependent DNA repair pathways emphasizes its potential as a promising therapeutic target for cancer treatment. Targeting SPIN1 could potentially enhance the efficacy of DNA repair-based therapies and overcome chemoresistance in cancer patients. Notably, several small molecule inhibitors targeting SPIN1 have already been developed. These include compounds like MS31 and A366, which block the interaction between SPIN1 and H3K4me3 (Fagan et al, 2019; Luise et al, 2021), as well as bivalent inhibitors, such as EML631 and VinSpinIn, which simultaneously interact with SPIN1's Tudor

domain I and II (Bae et al, 2017b; Fagan et al, 2019). Based on our findings, it is necessary to conduct further investigations and screenings to identify potent SPIN1 inhibitors that selectively disrupt the interaction between SPIN1 and H3K9me3 in our next study. Moreover, additional research is needed to elucidate the more precise molecular mechanisms underlying SPIN1's involvement in HR repair.

In conclusion, our study reveals a novel role for SPIN1 in coordinating the activation of H3K9me3-dependent DNA repair pathways. These findings highlight the critical involvement of SPIN1 in DNA damage repair and its potential contribution to cancer chemoresistance. Future investigations should focus on fully understanding its therapeutic potential and clinical implications of targeting SPIN1. It is crucial to explore strategies such as inhibiting the binding of SPIN1 to H3K9me3 or targeting SPIN1 itself to suppress DNA damage repair, thereby sensitizing cancer cells with high SPIN1 expression to DNA-damaging drugs.

# Methods

## Plasmids and siRNA

Human full-length SPIN1 (1–262) cDNA and its mutants (△1–50, △51–125, △125–190, △190–262) were cloned into the pEGFP-N1 or SFB (S-FLAG-SBP-tagged) vectors. The primers used are listed in Table EV1. Human full-length SFB-Tip60 was purchased from GenScript. The two siRNA sequences targeting SPIN1 are: (1) 5′UTR-5′-GGAUUAACCAGAACACUAAdTdT-3′; (2) CDS-5′-GCAAAGCAGUGGAACAUAUdTdT-3′. The transfection of plasmid was performed using Lipofectamine 2000 (Invitrogen) according to the manufacturer's instruction.

## Antibodies

Antibodies used in this study include the following: anti-SPIN1 (Proteintech, #12105-1-AP), anti-H3K9me3 (Cell Signaling Technology, #13969), anti-H3 (Cell Signaling Technology, #3638), anti-GAPDH (Cell Signaling Technology, #2118), anti-FLAG (Sigma, #F1804), anti-FLAG (Cell Signaling Technology, #2368), anti-anti-PAR (R&D Systems, #4335-mc-100), anti-GFP (Proteintech, #50430-2-AP), anti-γH2AX (Cell Signaling Technology, #2577),

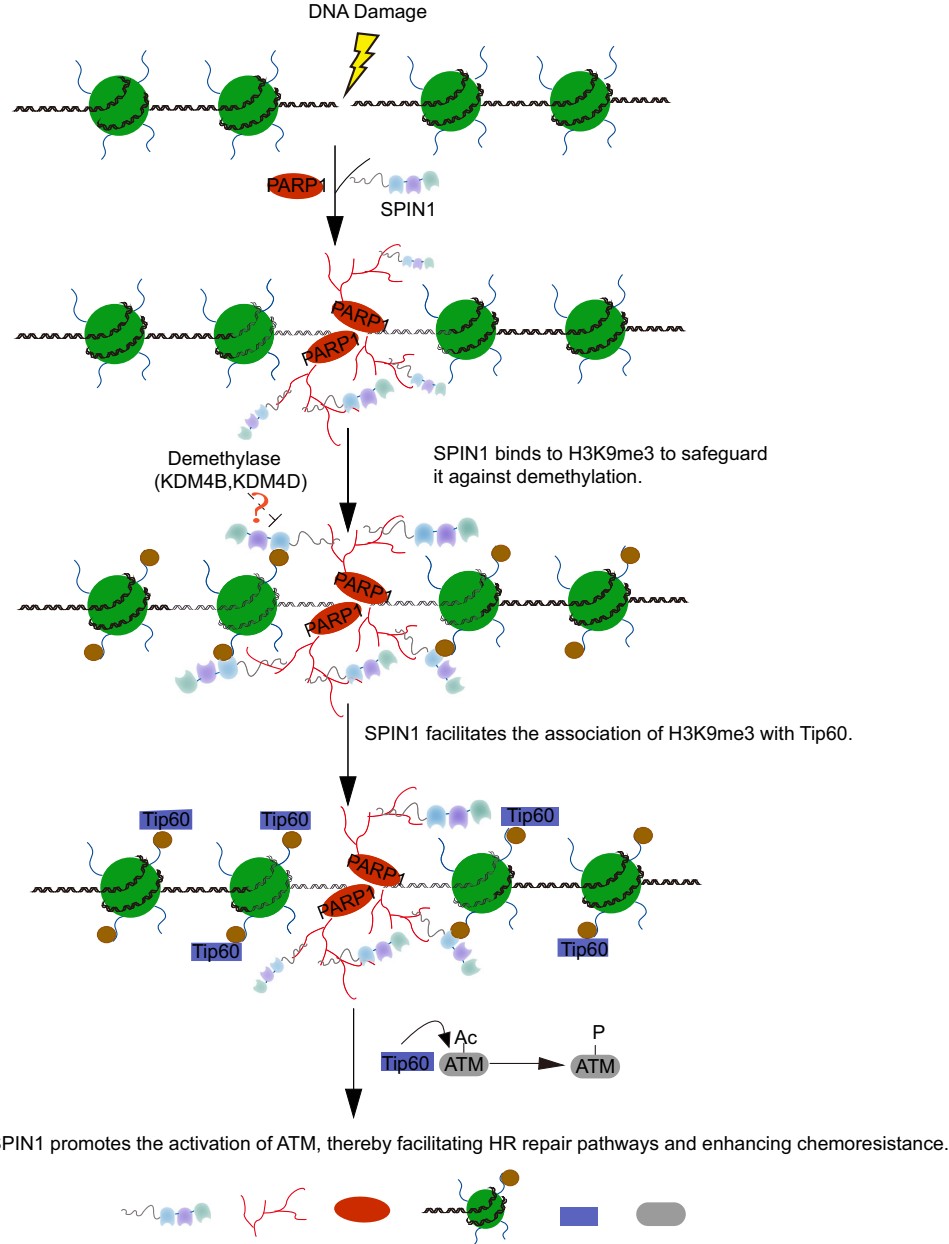

DNA Damage

PARP1
SPIN1

Demethylase
(KDM4B,KDM4D)

SPIN1 binds to H3K9me3 to safeguard
it against demethylation.

SPIN1 facilitates the association of H3K9me3 with Tip60.

SPIN1 promotes the activation of ATM, thereby facilitating HR repair pathways and enhancing chemoresistance.

SPIN1   PAR   PARP1   H3K9me3   Tip60   ATM

**Figure 6. A proposed working model of SPIN1 in DNA damage repair.**

The recruitment of SPIN1 by PAR to the site of DNA damage facilitates its interaction with H3K9me3, which in turn promotes the binding of H3K9me3 to Tip60. This leads to enhanced ATM activity and increases the capacity for DNA damage repair.

anti-BRCA1 (Santa Cruz Biotechnology, #sc-6954), anti-RAD51 (Abcam, #ab133534), anti-53bp1 (Abcam, #ab36823), anti-P-ATM (Cell Signaling Technology, #13050), anti-ATM (GeneTex, #GTX70103), anti-CHK2 (Proteintech, #13954-1-AP), anti-P-CHK2 (Proteintech, #81740-1-RR), anti-H3K4me3 (Cell Signaling Technology, #9751), anti-ATR (Cell Signaling Technology, #2790), anti-P-ATR (Cell Signaling Technology, #2853), anti-Tip60 (Proteintech, #10827-1-AP), anti-Pan Acetylation (Proteintech, #66289-1-Ig). HRP conjugated secondary antibodies were purchased from Cell Signaling Technology (#7076, #7074, #93702). Fluorescent

secondary antibodies were purchased from Invitrogen (#A-11034, #A-11032, #A-11001).

## Cell culture and stable cell lines

U2OS, HEK293T, HeLa, and SGC7901 cells were cultured in DMEM (HyClone) medium with 10% fetal bovine serum (HyClone), penicillin (100 U/ml), streptomycin (100 g/ml) and cultivated at 37 °C in 5% $CO_2$ (v/v). To generate SPIN1 knockdown cell lines, SGC7901 cells were transfected with vector (pLKO.1-puro) or pLKO.1-shSPIN1.

Transfected cells were plated at a low density in 5.0 µg/ml puromycin. Individual clones were isolated and validated by western blotting using anti-SPIN1 antibody.

## Laser microirradiation

U2OS cells were grown on 35 mm glass-bottom dishes (Nest, China). Laser microirradiation was performed on OLYMPUS FV3000 Confocal Laser Scanning Microscope with a Micropoint® Laser Illumination. The laser output was set to 70% using the 405 nm, which can reproducibly give a focused stripe by immunofluorescence staining with indicated antibodies. For time-lapse microscopic analysis, firstly, cells were transfected with corresponding plasmids. Then, green fluorescent protein (GFP) positive cells were subjected to microirradiation. The GFP strips were recorded from 30 s to 10 min and images were taken by the same microscope with the CellSens software (Olympus). GFP fluorescence at the laser line was converted into numerical value (relative fluorescence intensity) using Image J software. The error bars represent the SD.

## Immunofluorescence

To examine ionizing radiation-induced foci, U2OS cells were cultured on coverslips and treated with control or SPIN1 siRNA. After 72 h, cells were irradiated with 10 Gy IR and recovered for 4 h. Cells were fixed in 4% paraformaldehyde for 10 min and permeabilized with 0.5% Triton X-100 in phosphate-buffered saline (PBS) for 10 min at room temperature. Samples were blocked with blocking buffer (8% goat serum in PBS) for 20 min and then incubated with indicated antibodies, followed by incubation with fluorescently labeled secondary antibodies for 1 h at room temperature. Stained the nuclei with DAPI and count the number of foci in at least 50 cells/samples.

## Chromatin fractions extraction

The cells were harvested and washed twice with phosphate-buffered saline (PBS). Subsequently, the cells were lysed directly in NETN300 lysis buffer (20 mM Tris-HCl, pH 8.0, 300 mM NaCl, 1 mM EDTA, and 1% NP-40). The insoluble fractions were then digested using 0.5 U/µl of Benzonase. The chromatin fractions were extracted and then subjected to either electrophoresis or immunoprecipitation, followed by Western blot or dot blot analysis.

## Cell lysis, immunoprecipitation (IP), and Western blotting

The HEK293T cells were harvested after relevant treatment and washed twice with phosphate-buffered saline (PBS). Subsequently, the cell pellets were resuspended in the NETN-300 buffer (20 mM Tris-HCl, pH 8.0, 300 mM NaCl, 1 mM EDTA and 0.5% NP-40). For immunoprecipitation, the lysates were used with indicated antibodies and incubated with Pierce Protein G Agarose (Thermo) or High Capacity Streptavidin Agarose (Thermo Fisher Scientific, #20359) for 2 h at 4 °C. The beads were washed with NETN-100 buffer (20 mM Tris-HCl, pH 8.0, 100 mM NaCl, 1 mM EDTA, and 1% NP-40) for three times. After washing three times with NETN100, the samples were mixed with protein loading buffers

(25 mM Tris-HCl (PH 6.8), 10% SDS, 5 mg/ml Bromophenol blue, 50% Glycerine, 10% β-Mercaptoethanol) and denatured at 98 °C for 5 min. The protein lysates were separated using SDS-polyacrylamide gel electrophoresis and transferred to PVDF membranes (Merck Millipore, #IPVH00010). After blocking with 5% non-fat milk for 30 min at room temperature, the membranes were incubated with specific primary antibodies overnight at 4 °C and followed by incubated with secondary antibody for 1 h at room temperature. Finally, protein bands were visualized by enhanced chemiluminescent detection kit (Thermo, #34577) and the Bio-Rad ChemiDoc MP XRS+ Imaging System (Bio-Rad).

## GST fusion protein expression and Pull-down assay

Both full-length SPIN1 and the mutant △1–50 were expressed as GST-tagged recombinant proteins in *Escherichia coli* BL21(DE3) cells, following the previously described method for expressing GST fusion proteins. (Zhao et al, 2007). And PAR was a gift from Dr. Xiuhua Liu (Hebei University). Purified GST fusion proteins were incubated with PAR and Glutathione agarose beads (GE Healthcare) for 2 h at 4 °C. After washing with NETN-100 buffer, samples were boiled in SDS sample buffer and eluates were analyzed by Western blotting or dot blotting with indicated antibody.

## The biosynthesis and purification of PAR

The biosynthesis and purification of PAR polymer were carried out as described (Barkauskaite et al, 2013; Kam et al, 2018; Lin et al, 2018; Tan et al, 2012). Briefly, a 20 ml incubation buffer containing 100 mM Tris-HCl pH 7.8, 10 mM $MgCl_2$, 1 mM $NAD^+$, 1 mM DTT, 60 µg calf thymus histone (Sigma, #H5505), 50 µg octameric 'activator' oligonucleotide GGAATTCC, and 1.2 mg PARP1. The mixture was incubated at 30 °C for 60 min and stopped by adding 20 ml of ice-cold 20% TCA. DNase I was used to remove Oligo DNA and proteinase K was employed to digest proteins. After extraction with phenol-chloroform-isoamyl alcohol, PAR was precipitated with ethanol overnight. The PAR polymer was then dissolved and finally purified by anion exchange chromatography.

## Dot blot

After washing the agarose beads with NETN100 buffers, the samples were mixed with 0.1% SDS at 98 °C for 10 min and spotted onto a nitrocellulose membrane (GE Healthcare). After drying at room temperature, placed the nitrocellulose membrane into the UVP HL-2000 HybriLinker for auto-crosslink program twice, followed by western blotting analysis using the indicated antibodies.

## Comet assays

We performed single-cell gel electrophoresis comet analysis of DNA double-strand breaks under neutral conditions. HEK293T cells were treated with or without 10 Gy, after incubating for indicated time in fresh culture medium at 37 °C. Cells were harvested at $1 \times 10^5$ cells/mL in PBS and combined with 1% agarose with low melting point (LMP) at a 1:10 ratio (v/v) and immediately pipetted onto slides. Then, the slides were immersed in the neutral lysis buffer (2% sarkosyl, 0.5 M EDTA, 0.5 mg/mL proteinase K, pH

8.0) overnight at 37 °C. After lysis, the slides were washed with the electrophoresis buffer (90 mM Tris-HCl at pH 8.5, 90 mM boric acid, 2 mM EDTA), and analyzed by electrophoresis at 25 V for 40 min (0.6 V/cm) at 4 °C. Finally, the slides were stained in 10 µg/mL propidium iodide for 30 min in the dark. Images were taken with a fluorescence microscope (Olympus) and analyzed with the CometScore software.

## Colony formation assay

HEK293T cells were transfected with indicated plasmids or the indicated siRNA, and seeded into a 6-well plate at a density of 1000 cells/well. Cells were then exposed to ionizing radiation (0,1, 2 or 4 Gy). After 10 to 14 days, the live cells were washed with PBS, fixed in methanol for 5 min at room temperature, and stained with Giemsa. Then, the surviving cell fractions were calculated by comparing the numbers of colonies formed in the irradiated cultures with those in untreated control.

## Cell viability assay

For survival assay, cells were transfected with indicated plasmids or the indicated siRNA. After 48 h, cells were trypsinized, counted, and seeded into 96-well plates (Hela: $3 \times 10^3$; SGC7901: $5 \times 10^3$) in 100 µl medium with 10% FBS. Different concentration of Cisplatin (0, 2.5, 5, 7.5, 10 µM) or Olaparib (0, 25, 50, 75, 100 µM) was added for 48 h. After adding 10 µL of the CCK-8 reagent and incubated at 37 °C for 1 h, the absorbance values of treated cells were detected at 450 nm.

## Mouse xenograft analysis

Male 5-week-old NTG mice (NOD-scid IL2Rγ−/−) were purchased from SPF Biotechnology Co. (Beijing, China) and housed in pathogen-free animal laboratory for 1 week. And $2 \times 10^6$ control or SPIN1 knockdown SGC7901 cells were harvested and resuspended in 100 µl PBS and injected subcutaneously into the flanks of each mouse. Mice injected with the same cells were randomly allocated into four groups. We injected Cisplatin (3 mg/kg/two days) or Olaparib (50 mg/kg/day) for 14 days after the tumor forms for 10 days. Tumors' size was measured by a caliper every 2 days. The tumor volume was calculated using the following formula: tumor volume (mm³) = length*width*width/2. Following the surgical excision of the tumors, the specimens underwent photography, weighing, and documentation of tumor volumes. This study received support from the Animal Experiment Ethics Committee of Hebei University. The animal protocol described below has been reviewed and approved by the Animal Ethical and Welfare Committee (AEWC IACUC-2020XG013).

## NHEJ/HR in vivo reporter assays

HEK293T cells were inoculated at a density of $1 \times 10^6$ per 35 mm diameter dish. Cells then were transfected with DR-GFP, I-SceI, and m-Cherry plasmids or the NHEJ-GFP plasmid cleaved by *Hin*d III. After 6–8 h, the transfection medium was replaced with fresh cell culture medium. Thirty-six hours after transfection, cells were trypsinized, collected, and suspended in a PBS solution for subsequent flow cytometry analysis. The results represent the mean value of

triplicated replications in each experiment. In this experiment, m-Cherry was used to normalize for transfection efficiency.

## Flow cytometry analysis of cell cycle

For the cell cycle assay, HEK293T or HeLa cells were collected 72 h after transfection. The cells were washed with phosphate-buffered saline and fixed with 75% ethanol at −20 °C overnight. After centrifugation at $500 \times g$ for 3 min, cells were resuspended in PBS containing 1 mg/ml RNaseA and incubated for 30 min. Then, the cells were stained with propidium iodide (50 g/ml) at 37 °C for 30 min. The harvested cells were analyzed by flow cytometry based on the protocol provided by manufacturer.

## Statistical analysis

Statistical analysis was performed using GraphPad Prism 7.0. *P* values were analyzed by two-sided Student's t test or one-way ANOVA followed by the Tukey Kramer test. Data are plotted with error bars representing the standard error of the mean (SEM) or standard deviation of the mean (SD), as indicated in Figure legends. Statistical significance levels are denoted as follows: *$P < 0.05$; **$P < 0.01$; ***$P < 0.001$; ns, not significant.

# Data availability

This study includes no data deposited in external repositories.

The source data of this paper are collected in the following database record: biostudies:S-SCDT-10_1038-S44319-024-00219-1.

# Peer review information

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

## Acknowledgements

We would like to thank the research center for DNA damage repair of Hebei University for providing us technical assistance. This work was supported by National Natural Science Foundation of China (No. 32071277, No. 32171295), Natural Science Foundation of Hebei Province (No. C2021201012), Science and Technology Program of Hebei (No. 216Z2602G), Interdisciplinary Research Program of Natural Science of Hebei University (DXK202006).

## Author contributions

**Yukun Wang**: Resources; Formal analysis; Visualization; Methodology; Writing—original draft; Writing—review and editing. **Mengyao Li**: Data curation; Visualization. **Yuhan Chen**: Visualization. **Yuhan Jiang**: Validation. **Ziyu Zhang**: Validation. **Zhenzhen Yan**: Supervision; Methodology. **Xiuhua Liu**: Supervision; Project administration; Writing—review and editing. **Chen Wu**: Formal analysis; Supervision; Funding acquisition; Methodology; Writing—original draft; Writing—review and editing.

Source data underlying figure panels in this paper may have individual authorship assigned. Where available, figure panel/source data authorship is listed in the following database record: biostudies:S-SCDT-10_1038-S44319-024-00219-1.

## Disclosure and competing interests statement

The authors declare no competing interests.

*EMBO reports*                                                                 *Yukun Wang et al*

# Expanded View Figures

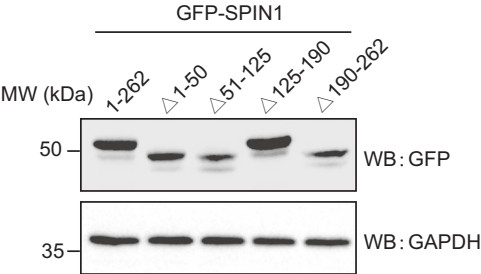

**Figure EV1.   Protein expressions of GFP-tagged SPIN1 mutants were examined by Western blot analysis.**

The indicated mutants of GFP-tagged SPIN1 were transfected to U2OS cells, and Western blot was performed to detect the protein expression levels. Source data are available online for this figure.

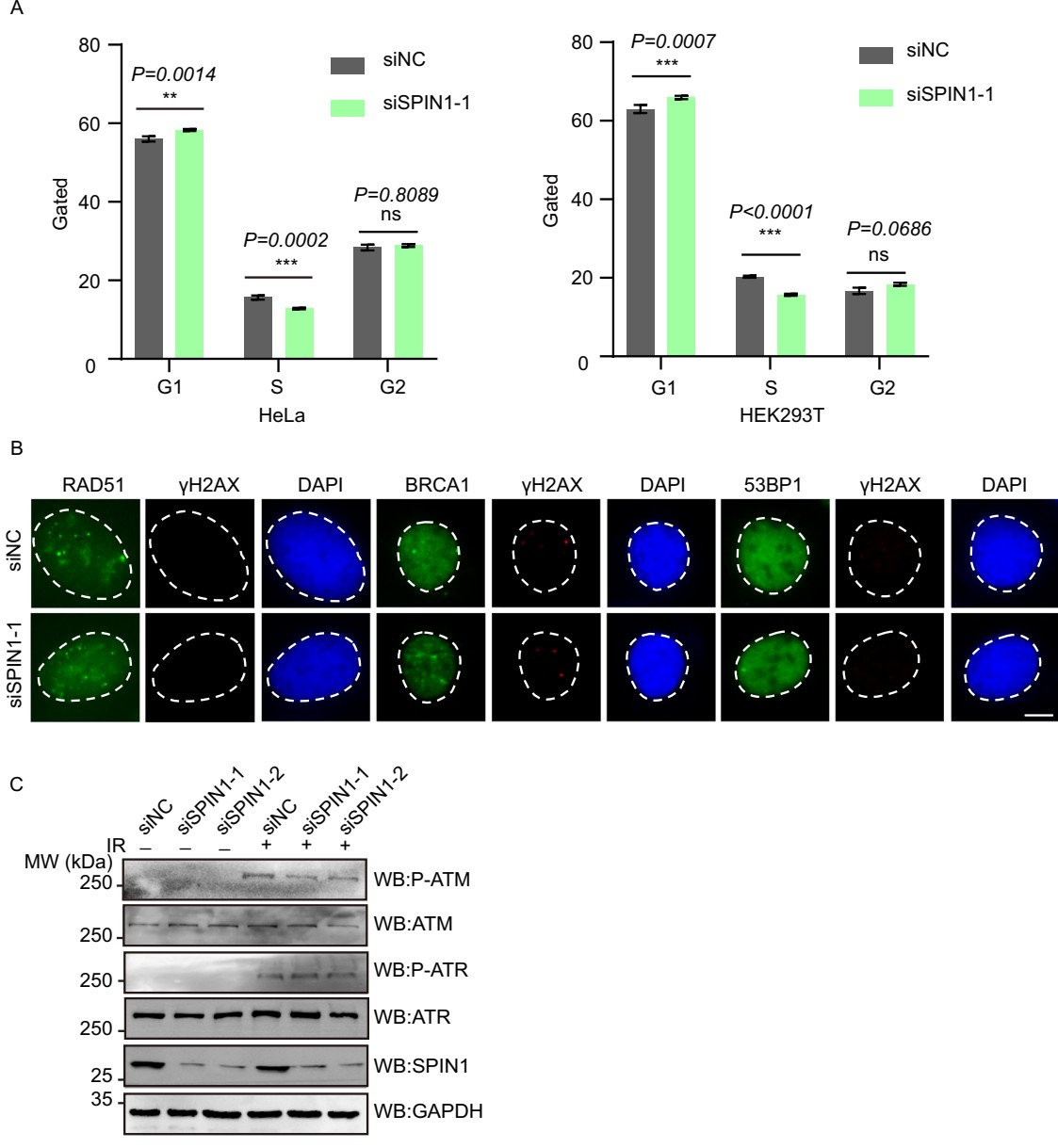

**Figure EV2.** (A) Knockdown of SPIN1 resulted in a slight G1/S shift in HeLa and HEK293T cells. HeLa and HEK293T cells were transfected with the indicated siRNAs and subsequently analyzed by flow cytometry. Three independent experiments were performed. Statistical significance was determined using one-way ANOVA followed by the Tukey Kramer test. The data are represented as the mean ± SD. **$P < 0.01$; ***$P < 0.001$; ns, not significant. (B) Knockdown of SPIN1 did not induce DNA damage. U2OS cells were transfected with the indicated siRNAs, and the formation of γH2AX foci was examined by immunofluorescent staining. Scale bar = 10 μm. (C) Knockdown of SPIN1 resulted in a decrease in the levels of phosphorylated ATM (P-ATM), but not phosphorylated ATR (P-ATR). HEK293T cells were transfected with the indicated siRNAs and either treated or untreated with 10 Gy of IR. Total cell lysates were collected and subjected to immunoblotting using the indicated antibodies. Source data are available online for this figure.

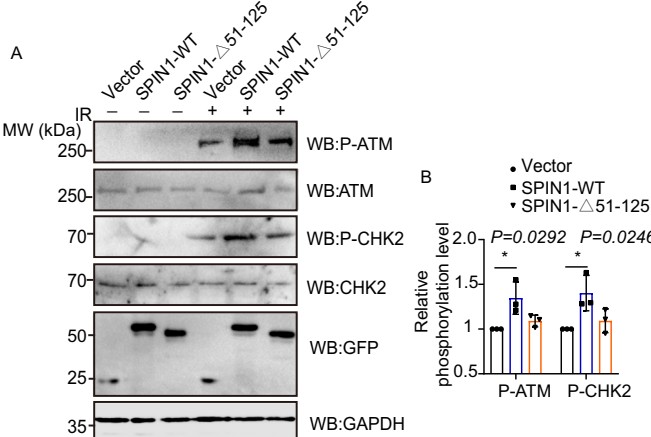

**Figure EV3.** (A) Overexpression of SPIN1-WT, but not the 51–125 amino acid deletion mutant, promoted the activation of ATM upon DNA damage. HEK293T cells expressing the vector, SFB-SPIN1-WT or SFB-SPIN1-△51–125 were treated or untreated with 10 Gy of IR. Total cell lysates were harvested and subjected to immunoblotting with the indicated antibodies. (B) Quantitative statistical analysis was performed on the phosphorylation levels of ATM and CHK2 from three independent biological replicates. Statistical significance was determined using the Student's t-test. Data are presented as mean ± SD. *P < 0.05. Source data are available online for this figure.

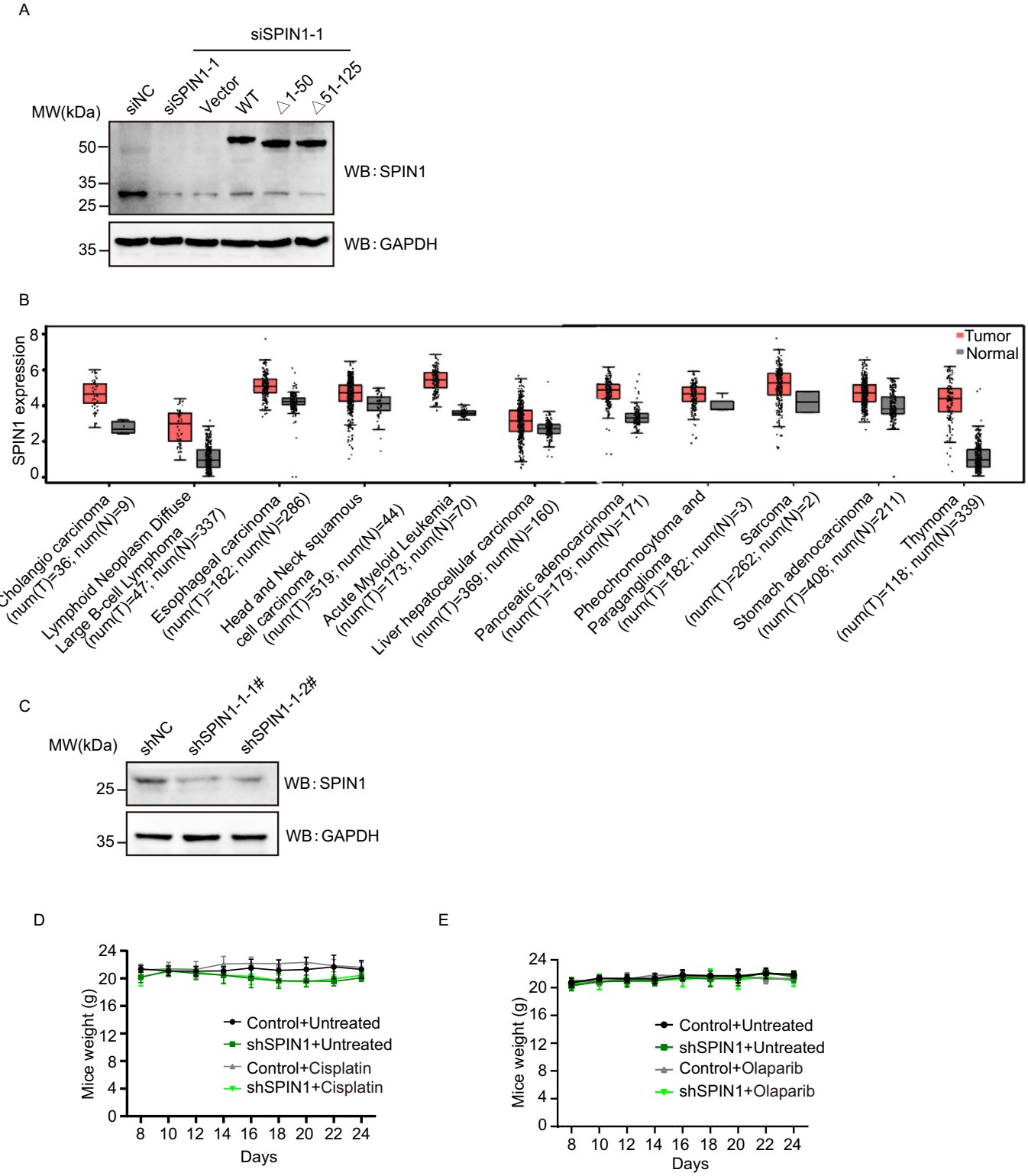

**Figure EV4.** (A) Protein expressions of GFP-tagged SPIN1 mutants were examined by Western blot analysis. The indicated mutants of GFP-tagged SPIN1 were transfected, and Western blot was performed to detect the protein expression levels. (B) The comparisons of SPIN1 expression between tumor and normal tissues were conducted using data from the GEPIA database(http://gepia.cancer-pku.cn/). The horizontal line within each box represents the median, and the box boundaries are defined by the 25th and 75th percentiles. The whiskers extend to the minimum and maximum values. (C) The protein levels of SPIN1 were analyzed by Western blot in stable shSPIN1 or shNCSGC7901 cells. (D, E) The body weight of the mice was measured during the treatment, and no significant weight loss was observed. The graphs represent the mean ± SD. $n = 6$/group. Source data are available online for this figure.

