## [Peer Review File · EMBO Reports]

SPIN1 facilitates chemoresistance and HR repair by promoting Tip60 binding to H3K9me3

Yukun Wang, Mengyao Li, Yuhua Chen, Yuhua Jiang, Ziyu Zhang, Zhenzhen Yan, Xiuhua Liu, and Chen Wu

Corresponding author(s): Xiuhua Liu (xiuhualiu@hbu.edu.cn, liuxiuhua_2004@163.com), Chen Wu (wuchen@hbu.edu.cn)

Review Timeline:

Submission Date:	17th Jan 24
Editorial Decision:	22nd Feb 24
Revision Received:	6th Jun 24
Editorial Decision:	3rd Jul 24
Revision Received:	8th Jul 24
Accepted:	18th Jul 24

Editor: Esther Schnapp

Transaction Report:

Dear Prof. Wu,

Thank you for the submission of your manuscript to EMBO reports. We have now received the full set of referee reports as well as additional comments from referee 3, which are all pasted below.

As you will see, the referees acknowledge that the findings are potentially interesting. However, referee 3 points out that some more mechanistic insight should be provided. I think all comments and suggestions are good and should be addressed. Please let me know in case you disagree, and we can discuss the exact revision requirements further, also in a video chat, if you like.

I would thus like to invite you to revise your manuscript with the understanding that the referee concerns must be fully addressed and their suggestions taken on board. Please address all referee concerns in a complete point-by-point response. Acceptance of the manuscript will depend on a positive outcome of a second round of review. It is EMBO reports policy to allow a single round of major revision only and acceptance or rejection of the manuscript will therefore depend on the completeness of your responses included in the next, final version of the manuscript.

We realize that it is difficult to revise to a specific deadline. In the interest of protecting the conceptual advance provided by the work, we recommend a revision within 3 months (24th May 2024). Please discuss the revision progress ahead of this time with the editor if you require more time to complete the revisions.

- 1) A data availability section providing access to data deposited in public databases is missing. If you have not deposited any data, please add a sentence to the data availability section that explains that.
- 2) Your manuscript contains statistics and error bars based on $n=2$. Please use scatter blots in these cases. No statistics should be calculated if $n=2$.

5) a complete author checklist, which you can download from our author guidelines <https://www.embopress.org/page/journal/14693178/authorguide>. Please insert information in the checklist that is also reflected in the manuscript. The completed author checklist will also be part of the RPF.

6) Please note that all corresponding authors are required to supply an ORCID ID for their name upon submission of a revised manuscript (<https://orcid.org/>). Please find instructions on how to link your ORCID ID to your account in our manuscript tracking system in our Author guidelines <https://www.embopress.org/page/journal/14693178/authorguide#authorshipguidelines>

7) Before submitting your revision, primary datasets produced in this study need to be deposited in an appropriate public

database (see <https://www.embopress.org/page/journal/14693178/authorguide#datadeposition>). Please remember to provide a reviewer password if the datasets are not yet public. The accession numbers and database should be listed in a formal "Data Availability" section placed after Materials & Method (see also <https://www.embopress.org/page/journal/14693178/authorguide#datadeposition>). Please note that the Data Availability Section is restricted to new primary data that are part of this study. * Note - All links should resolve to a page where the data can be accessed. *

10) Regarding data quantification (see Figure Legends:

<https://www.embopress.org/page/journal/14693178/authorguide#figureformat>)

- the name of the statistical test used to generate error bars and P values,
- the number (n) of independent experiments (please specify technical or biological replicates) underlying each data point,
- the nature of the bars and error bars (s.d., s.e.m.),
- If the data are obtained from $n < 3$, please use scatter blots showing the individual data points.

I look forward to seeing a revised form of your manuscript when it is ready.

Yours sincerely,

Referee #1:

The manuscript entitled 'SPIN1 enhances H3K9me3/Tip60 binding to facilitate HR repair pathway and chemoresistance' focuses on the SPIN1 protein and its role in repair. SPIN1 is a transcription factor with a disordered region and three tudor domains. These are known to bind modified histones. SPIN1 is also overexpressed in some cancers, and it has known functions in regulating transcription and mediating cell cycle progression. Here, the authors demonstrate a role for SPIN1 in mediating repair by homologous recombination. They provide evidence that SPIN1 is recruited to sites of DNA damage via interaction between the N-terminal disordered region and PAR. They further show that SPIN1 impacts on H3K9me3, Tip60 recruitment, ATM activation, and cell survival following irradiation. Overall, this is a nice study that extends the understanding of DNA damage responses. However, in places, the conclusions are overstated, and some additional controls are required as outlined below.

Major issues:

1. In Figure 1, the analysis of the truncations is not as conclusive as stated in the text. Specifically, the background signal in the GFP-tagged N-terminal truncation (del1-50) is much higher than the other constructs and appears to be interfering with the quantification of recruitment. A damage-associated signal is clearly visible in the images provided, but the quantification in the right panel does not reflect this. Given the data provided in Figure 2, it is reasonable to conclude that the recruitment of this construct to damage is somewhat impaired, but it is not as substantial as the graph implies. The statements should be modified in the text and the graph should be corrected to account for the technical issues.
2. In Figures 2 and 4, the authors provide a series of immunoprecipitations of chromatin bound proteins (including histones), but according to the information provided in the methods section, there is no step in which the chromatin is fragmented or digested, suggesting these interactions could be indirect. There is an experiment using recombinant SPIN1 and PAR, suggesting a direct interaction of these two molecules at least. Nevertheless, at least the key IP assays should be performed using sonication or nuclease digestion of the chromatin prior to pull-down.
3. In Figure 3 there is no control for the potential impact of SPIN1 depletion on cell cycle progression. All the data in Figure 3 could be explained by a shift in cell cycle (if siSPIN1 leads to an increase in G1 cells and decrease in G2 cells) rather than a defect in HR. Therefore, the cell cycle profile of SPIN1 depleted cells alongside the siRNA negative control cells should be provided.
4. In Figure 5A, the conclusions are overstated. The truncations are not all expressed to the same level as full-length SPIN1 according to the data provided in the Supplemental Figures. Therefore, the failure of the mutants to rescue could be due to lower expression rather than lack of specific domains. This should be mentioned as a possibility in the text and the conclusion stated much more cautiously.
5. In Figure 5, the survival following damage of the U2OS cell line treated with siSPIN1 is not tested, despite this line being used in all the other experiments to elucidate the pathway. This should be added.
6. In Figure 5C and 5D, the authors don't comment on the surprising impact of SPIN1 depletion on Olaparib sensitivity. If (as shown in Figure 2B), SPIN1 is not recruited to damaged DNA in Olaparib treated cells, then there should be no impact of SPIN1 expression or depletion on the cellular response to Olaparib. The data in Figure 5 therefore implies that there is some residual SPIN1 recruitment even after Olaparib treatment or that there is another separate function for SPIN1 in mediating survival after Olaparib treatment. This issue should be covered in the text.

Minor issues:

- The authors could consider a figure showing the levels of SPIN1 overexpression in cancers (using publicly available TCGA data).
- The number of gH2AX foci should also be provided in Figure 3 to compare with the other quantification data. In addition, images of untreated cells should be provided (in Supplementary Data) to show that the siRNA treatments do not cause any damage on their own.
- It would be informative to have analysis of ATR activation in parallel with the ATM data.
- There are a number of typos in the manuscript that should be corrected. For example, in Figure 4F, one of the panels is labelled 'FALG' rather than 'FLAG'. Also, there are problems with the reference formatting. For example, Jin et al is missing the publication year (2021) and the capitalization of journal titles is inconsistent.
- There was a paper showing that SPIN-DOC binds to PARP1, which could represent an interesting regulatory mechanism (PMID: 34737271). It would be good to cite this reference and discuss implications in the text.

Referee #2:

In this study, the authors have demonstrated involvement of SPIN1 in HR-mediated double strand break repair and show evidence supporting SPIN1 promotes association of Tip60 to H3K9me3 generated after DNA damage. They propose that SPIN1 association with H3K9me3 protects this mark against demethylase activity and serves to promote downstream activation of ATM and HR DNA damage repair. They identify SPIN1 as a potential target to increase sensitivity of cells to DNA damaging therapeutic agents such as ionizing radiation and chemotherapy. Overall the data shown generally supports their conclusions with modest effects of chemo- and radio-sensitization in the tested model.

Revisions suggested:

-Figure 1 should include labels to clearly delineate which constructs represent full-length vs. specific tudor domain deletions

-The authors suggest that SPIN1-H3K9me3-Tip60 form a complex promoting ATM activation and DNA damage repair and that SPIN1 is recruited by PAR binding, however it is unclear what form of PAR they are studying. Poly ADP ribosylation exists on multiple substrate types including PARP1 itself and histones. The paper requires better characterization of the PAR that was utilized in experiments shown in Figure 2. The methods section lists this reagent as a gift without analytics; characterization should be provided in the supplement to allow for better understanding of and reproduction of the experiment.

-Figure 2: pulldown assays should include input lanes

-Evidence for direct binding between SPIN1 and PAR and discussion of binding kinetics would be strengthened with orthogonal binding experiment data e.g. SPR, ITC; alternatively, would suggest that body and figure text should be rephrased to reflect co-IP experiments e.g. refer to increased pulldown efficiency instead of increased binding affinity as K_D values are not reported.

-Figure 3 should quantify the knockdowns as shown on WB in addition to flow cytometry

-Regarding Figure 3: text or legend should define the difference between SPIN1-1 and SPIN1-2 (the methods clarify that these two different siRNA sequences)

-Figures implying kinetics should fit data to a curve or plot quantification data separately e.g. Figure 4D, Figure 5B-D

-Further discussion of the role of related protein SPINDOC is lacking; although this related protein is briefly mentioned in the text, available structural data and previous work outlining the role of SPINDOC in DNA damage response should be cited, if not also included as part of the model in the final figure.

citations below:

<https://doi.org/10.1016/j.jmb.2023.168371>

doi: 10.1038/s41467-021-26588-y

-The authors do acknowledge that small molecule inhibitors of SPIN1 have been published and discuss future experiments with them, but if these are available to them presently, they would provide additional support. A suggested experiment would be treatment with SPIN1 inhibitors in a time course following DNA damaging treatment to understand the kinetics of SPIN1 mediated DNA damage repair.

Referee #3:

SPIN1 is known to be a reader of H3K9me3. Histone H3K9 methylation links DNA damage detection to activation of Tip60. Based on these background, in this study, authors revealed a new function of SPIN1 in binding to DNA damage sites. Firstly, they reported that SPIN1 is recruited to DNA lesions through its N-terminal disordered region that binds to PAR, and facilitates homologous recombination (HR)-mediated DNA damage repair. Secondly, SPIN1 maintains the stability of H3K9me3 and enhances the interaction between H3K9me3 and Tip60, leading to the activation of ATM and HR repair. Finally, authors demonstrate that SPIN1 contributes to chemoresistance by enhancing DNA repair. However, the detail mechanism underlying the epigenetic changes is totally unknown. In addition, some issues such as lack of appropriate controls or approaches preclude publication for EMBO reports.

Comments:

1. Authors state that SPIN1 preserves the presence of H3K9me3 at the site of damage, however, the exact mechanism is absent, although lysine demethylase KDM4B is mentioned. How does SPIN1 regulate KDM4B? Is KDM4B involved in the regulation of SPIN1 on H3K9me3?

2. Authors confirm that SPIN1 binds to PAR dependent of the N-terminus IDR. Authors also mentioned in the introduction that the first Tudor module of SPIN1 can recognize H3R8me2a or H3K9me3. Does SPIN1 bind to PAR or H3K9me3 through different domain? Does PAR inhibitor influence the interaction of SPIN1 and H3K9me3?

3. Authors stated that SPIN1 enhances chemoresistance by cell viability assay. Evidence in vivo is necessary for this conclusion. For example, both cisplatin and olaparib are used to treat the mice implanted of control or SPIN1-knockdown cells, and tumor sizes will be measured.

4. A one-way ANOVA with a non-parametric post hoc test or two-way ANOVA should be used to analyze the data rather than multiple t-tests. For example, Fig2B, Fig4D, Fig5. The detail of statistical method should be included in each figure legend.

Further comments from referee 3:

In this study, the authors reveal a new function of SPIN1 in binding to DNA damage sites, although the precise mechanism is not well described (comments1 and 2). Here, the important function of SPIN1 in DNA repair is to participate in chemoresistance. However, the related conclusion is overstated, and further experiments are required (comments3). In addition, there are some issues with the appropriateness of the statistical analysis that should be addressed (comments4).

Overall, as the standards of EMBOR, if the authors could reasonably respond to the comments by additional experiments or discussion, this manuscript can be in major revision.

Response to reviewers' comments:

We are very grateful for the constructive comments from the reviewers. Following the reviewers' suggestions, we have modified the manuscript. As listed below, we have point-by-point addressed all the concerns raised by reviewers.

Please note that page and line numbers that we mention here refer to our revised manuscript and not to the original submission. The changed and added texts and figures in the revised manuscript are highlighted in red font and underlined. Also, please note that the numbering of the figures in the revised manuscript is different from that in the original one.

Referee#1:

The manuscript entitled 'SPIN1 enhances H3K9me3/Tip60 binding to facilitate HR repair pathway and chemoresistance' focuses on the SPIN1 protein and its role in repair. SPIN1 is a transcription factor with a disordered region and three tudor domains. These are known to bind modified histones. SPIN1 is also overexpressed in some cancers, and it has known functions in regulating transcription and mediating cell cycle progression. Here, the authors demonstrate a role for SPIN1 in mediating repair by homologous recombination. They provide evidence that SPIN1 is recruited to sites of DNA damage via interaction between the N-terminal disordered region and PAR. They further show that SPIN1 impacts on H3K9me3, Tip60 recruitment, ATM activation, and cell survival following irradiation. Overall, this is a nice study that extends the understanding of DNA damage responses. However, in places, the conclusions are overstated, and some additional controls are required as outlined below.

Response: Thank you for the positive comments! The specific questions have been addressed below.

Major issues:

1. In Figure 1, the analysis of the truncations is not as conclusive as stated in the text. Specifically, the background signal in the GFP-tagged N-terminal truncation (del1-50) is much higher than the other constructs and appears to be interfering with the quantification of recruitment. A damage-associated signal is clearly visible in the images provided, but the quantification in the right panel does not reflect this. Given the data provided in Figure 2, it is reasonable to conclude that the recruitment of this construct to damage is somewhat impaired, but it is not as substantial as the graph implies. The statements should be modified in the text and the graph should be corrected to account for the technical issues.

Response: Thank you for your valuable comments. We apologize for the confusion

caused by the higher background signal observed in the GFP-tagged N-terminal truncation (Δ 1-50). Upon further examination, we found that the signal was not a laser stripe but rather an artifact. In our revised manuscript, we have replaced the images of the Δ 1-50 mutant with more representative figures and marked the laser stripes with white arrows(Fig. 1C). The revised figure clearly demonstrates that compared to SPIN1-WT, the Δ 1-50 mutant did not show a noticeable recruitment signal(Fig. 1C). This result was consistently observed in our pull-down assays, which demonstrated that the Δ 1-50 mutant failed to bind to PAR, while SPIN1's recruitment to DNA damage sites is dependent on PAR(Fig. 2G). This finding supports the crucial role of the IDR in SPIN1's recruitment to DNA damage sites. Given the presence of a cluster of Lysine and Arginine residues with positive charges at the N-terminal IDR of SPIN1, we speculate that the binding between SPIN1 and PAR may be mediated by electrostatic interaction.

Figure for referees not shown.

2. In Figures 2 and 4, the authors provide a series of immunoprecipitations of chromatin bound proteins (including histones), but according to the information provided in the methods section, there is no step in which the chromatin is fragmented or digested, suggesting these interactions could be indirect. There is an experiment using recombinant SPIN1 and PAR, suggesting a direct interaction of these two molecules at least. Nevertheless, at least the key IP assays should be performed using sonication or nuclease digestion of the chromatin prior to pull-down.

Response: Thank you for your suggestion. We sincerely apologize for any confusion caused by the lack of detailed description in our Materials and Methods section. In our original manuscript, we did use 0.5 U/ μ l Benzonase during the chromatin fractions extraction process to ensure thorough digestion of the chromatin. However, we inadvertently omitted a detailed description regarding this step in our Materials and Methods section. To address this issue, we have included the procedure for chromatin fractions extraction in our revised manuscript, emphasizing the use of nuclease for chromatin digestion (Page 18, Line 4-10).

In Figure 2G, we performed a pull-down assay using recombinant GST-SPIN1 purified from *E.coli* BL21(DE3) and protein-free PAR prepared through enzymatic synthesis. This result indicated a direct interaction between SPIN1 and PAR. To further strengthen the evidence for direct binding between SPIN1 and PAR, additional binding experiments such as SPR and ITC could be employed. We acknowledge the limitations of the method utilized in our study and emphasize the need for further research to obtain a more comprehensive understanding of the binding kinetics between SPIN1 and PAR(Page 13, Line 1-4).

3. In Figure 3 there is no control for the potential impact of SPIN1 depletion on cell cycle progression. All the data in Figure 3 could be explained by a shift in cell cycle (if siSPIN1 leads to an increase in G1 cells and decrease in G2 cells) rather than a defect in HR. Therefore, the cell cycle profile of SPIN1 depleted cells alongside the siRNA negative control cells should be provided.

Response: Thank you for your valuable comment. As suggested, we compared the cell cycle profiles between SPIN1-knockdown cells and the siRNA negative control cells. Depletion of SPIN1 did not cause an obvious G1/G2 shift (Fig. EV2A), which is consistent with previous studies (Yuan *et al*, 2008; Zhang *et al*, 2008; Zhao *et al*, 2007; Lv *et al*, 2020). The slight alteration in the cell cycle cannot fully account for the significant defect in HR observed in SPIN1-depleted cells. Moreover, our findings show that SPIN1 is recruited to DSBs sites and specifically promotes HR but not NHEJ, which further indicates the direct impact of SPIN1 on DSBs repair.

It has been shown that some DNA damage repair factors not only participate in DNA lesions repair but also play a key role in regulating the cell cycle. For example, BRCA1, a well-known HR factor, activates Chk1 kinase to initiate the G2/M checkpoint in response to DNA damage. Similarly, ATM not only initiates DSB responses but also activates signal transduction pathways to induce cell cycle arrest. We agree with you that SPIN1 also regulates the cell cycle in addition to its role in DNA damage repair. We have discussed this point and included the data on cell cycle profiles in the revised manuscript(Page 7, Line 19-25).

Figure for referees not shown.

4. In Figure 5A, the conclusions are overstated. The truncations are not all expressed to the same level as full-length SPIN1 according to the data provided in the Supplemental Figures. Therefore, the failure of the mutants to rescue could be due to lower expression rather than lack of specific domains. This should be mentioned as a possibility in the text and the conclusion stated much more cautiously.

Response: Thank you for your thorough review and constructive comment. In our original manuscript, we did observe a slight difference in the expression of the Δ 1-50 mutant compared to the full-length SPIN1. We have conducted additional experiments to address the concern raised and have made the following revisions. Firstly, as suggested in Comment #5, we have included the siSPIN1 group in our study. Additionally, we repeated the experiment depicted in Figure 5A to ensure that the truncations (Δ 1-50 and Δ 51-125) were expressed at similar levels as full-length SPIN1 (Fig. EV4A). Our findings indicate that the reintroduction of wild-type SPIN1, but not the Δ 1-50 or Δ 51-125 mutants, effectively restores the deficiency in DNA repair, indicating the vital role of intact SPIN1 in DSB repair.

Figure for referees not shown.

5. In Figure 5, the survival following damage of the U2OS cell line treated with siSPIN1 is not tested, despite this line being used in all the other experiments to elucidate the pathway. This should be added.

Response: Thank you for your comment. We apologize for the oversight in our original version. As suggested, we have conducted additional experiments to include the siSPIN1 group as a control in the survival assay. The updated results, including the comet assay and colony-formation assay, have been incorporated into the revised Figure 5A and 5B, respectively.

Figure for referees not shown.

Figure for referees not shown.

6. In Figure 5C and 5D, the authors don't comment on the surprising impact of SPIN1 depletion on Olaparib sensitivity. If (as shown in Figure 2B), SPIN1 is not recruited to damaged DNA in Olaparib treated cells, then there should be no impact of SPIN1 expression or depletion on the cellular response to Olaparib. The data in Figure 5 therefore implies that there is some residual SPIN1 recruitment even after Olaparib treatment or that there is another separate function for SPIN1 in mediating survival after Olaparib treatment. This issue should be covered in the text.

Response: We appreciate your insightful comment. We agree with you that these results suggest there might be alternative mechanisms by which SPIN1 contributes to cell survival in the context of Olaparib treatment. Previous studies have elucidated various mechanisms by which SPIN1 promotes cell proliferation and tumor growth. For example, SPIN1 inhibits apoptosis and promotes cell proliferation by inactivating p53 and blocking the uL18-MDM2-p53 pathway in human cancer(Fang *et al*, 2018). Additionally, SPIN1 has been shown to promote cell proliferation in gastric cancer through the activation of the MDM2-p21-E2F1 signaling pathway(Lv *et al*, 2020). As suggested, we have included a possible explanation for our results. Moreover, we acknowledge that further investigations are necessary to explore the specific mechanisms involved(Page 11, Line 20-23). These future studies will help us gain a deeper understanding of how SPIN1 influences cell survival in the context of Olaparib treatment.

Minor issues:

- *The authors could consider a figure showing the levels of SPIN1 overexpression in cancers (using publicly available TCGA data).*

Response: Thank you for your suggestion. We have incorporated this suggestion into our revisions and included the requested figure in the revised manuscript(Fig. EV4B). We utilized the GEPIA (Gene Expression Profiling Interactive Analysis) web server (<http://gepia.cancer-pku.cn/>) to analyze the levels of SPIN1 expression in different cancers. This analysis is based on data from the TCGA and the GTEx projects. The figure illustrates the elevated expression levels of SPIN1 across various cancer types compared to their corresponding normal tissues(Page 11, Line 4-9).

Figure for referees not shown.

- *The number of γ H2AX foci should also be provided in Figure 3 to compare with the other quantification data. In addition, images of untreated cells should be provided (in Supplementary Data) to show that the siRNA treatments do not cause any damage on their own.*

Response: Thank you for your suggestion. As suggested, we carefully counted the number of γ H2AX foci and compared the results between siNC and siSPIN1 groups. However, we found no significant difference in the formation of γ H2AX foci (Fig. 3F-H and J).

Furthermore, we have included the requested images of untreated cells in the Supplementary data section (Fig. EV2B). These results demonstrate that in the absence of IR treatment, the siRNA treatments do not induce any damage.

Figure for referees not shown.

Figure for referees not shown.

- *It would be informative to have analysis of ATR activation in parallel with the ATM data.*

Response: Thank you for your valuable suggestion. As suggested, we have conducted additional analyses and included the assessment of both of ATR and ATM activation. Our results showed that knockdown of SPIN1 resulted in decreased levels of phosphorylated ATM (P-ATM), while there were no significant changes observed in phosphorylated ATR (P-ATR) following DNA damage (Fig. EV2C). These results indicate that SPIN1 primarily promotes the activation of ATM rather than ATR. Consequently, our study focused on elucidating the mechanism through which SPIN1 activates ATM.

Figure for referees not shown.

- *There are a number of typos in the manuscript that should be corrected. For example, in Figure 4F, one of the panels is labelled 'FALG' rather than 'FLAG'. Also, there are problems with the reference formatting. For example, Jin et al is missing the publication year (2021) and the capitalization of journal titles is inconsistent.*

Response: Thank you for your careful review of our manuscript. We sincerely apologize for the typos and formatting errors in the manuscript. We have made the following corrections in the revised version:

- 1) Figure 4F: We have corrected the labeling of the panel from 'FALG' to 'FLAG'.
- 2) Reference formatting: We have addressed the issues with reference formatting throughout the manuscript. Specifically, we have added the publication year (2021) to the citation of Jin *et al*. Moreover, we have ensured consistent capitalization of journal titles to enhance the overall clarity and professionalism of the manuscript.
- 3) We have conducted a comprehensive review of the entire manuscript to identify and correct any other potential errors. For example, we have corrected the spelling 'olaparib' to 'Olaparib' and ' γ -H2AX' to ' γ H2AX'. Furthermore, all the revisions made during this process have been tracked.

- *There was a paper showing that SPIN-DOC binds to PARP1, which could represent an interesting regulatory mechanism (PMID: 34737271). It would be good to cite this reference and discuss implications in the text.*

Response: Thank you for your valuable suggestion regarding the papers on SPINDOC. As suggested, we have incorporated the citation of this reference in our revised version. SPINDOC, also known as C11orf84, is closely associated with SPIN1. Previous studies have reported two distinct roles of SPINDOC in regulating SPIN1. Firstly, it regulates the transcriptional activator activity of SPIN1 by inhibiting its histone methyl-binding ability, thereby repressing the expression of SPIN1-regulated genes and the SPIN1-mediated activation of the Wnt signaling pathway (Bae *et al*, 2017). Secondly, as recommended by the reviewer, the study (PMID: 34737271) highlights an alternative role of SPINDOC that is independent of SPIN1. This study demonstrates that SPINDOC interacts with PARP1 and facilitates its ADP-

ribosyltransferase activity in response to DNA damage, indicating that SPINDOC has a SPIN1-independent role in regulating PARP1-mediated PARylation and the DNA damage response(Yang *et al*, 2021).

In 2021, Du *et al*. determined the crystal structure of the SPIN1 (aa:50-262)-C11orf84 (aa:253-295)-H3K4me3K9me3 (A1-G12) ternary complex (PDB code: 7CNA). They revealed that the C11orf84(SPINDOC) fragment binds to the Tudor 3 domain of SPIN1, while the methyl histone peptide spans across the Tudor 1 and 2 domains. Furthermore, Du *et al*. showed that C11orf84 binding stabilizes SPIN1 and enhances its association with the bivalent H3K4me3K9me3 mark, thereby facilitating SPIN1's activation in rRNA transcription(Du *et al*, 2021). Recently, Zhao *et al*. resolved the crystal structures of SPIN1 (aa:50-262) bound to SPINDOCpep3 (aa:256-281) (PDB code: 7E9M) and SPINDOCpep2 (aa:228-239) (PDB code: 7EA1), respectively (Zhao *et al*, 2024). The SPIN1-SPINDOCpep3 complex revealed that SPINDOCpep3 (aa:256-281) interacts with the Tudor-like 3 domain of SPIN1, consistent with the structural analysis reported by Du *et al*. Furthermore, they found that two neighboring K/R-rich motifs (SPINDOCpep1 and SPINDOCpep2) bind to the acidic surface of SPIN1's Tudor-like 2 domain, thereby attenuating SPIN1's binding to the methylated H3 and its transcriptional co-activator activity(Zhao *et al*, 2024). As a result, these studies present contradictory results regarding the regulatory role of SPINDOC on SPIN1, particularly in enhancing SPIN1's association with methylated H3.

In our revised manuscript, we have included citations of the available structural data and previous studies that highlight the involvement of SPINDOC in the DNA damage response. Additionally, considering that SPINDOC binds PARP1 to facilitate PARylation and DNA damage response independent of SPIN1(Yang *et al*, 2021), we have included a discussion on the potential implications of SPINDOC binding to SPIN1 in the context of SPIN1's involvement in DNA damage repair(Page14, Line 20-29). However, further research is needed to explore the precise regulatory mechanism of SPINDOC in SPIN1's involvement in DNA damage repair.

Reference:

- Bae N, Gao M, Li X, Premkumar T, Sbardella G, Chen J, Bedford MT (2017) A transcriptional coregulator, SPIN1, attenuates the coactivator activity of Spindlin1. *The Journal of biological chemistry* 292(51): 20808-20817.
- Du Y, Yan Y, Xie S, Huang H, Wang X, Ng RK, Zhou M-M, Qian C (2021) Structural mechanism of bivalent histone H3K4me3K9me3 recognition by the Spindlin1/C11orf84 complex in rRNA transcription activation. *Nature communications* 12(1):949.
- Fang Z, Cao B, Liao JM, Deng J, Plummer KD, Liao P, Liu T, Zhang W, Zhang K, Li L, Margolin D, Zeng SX, Xiong J, Lu H (2018) SPIN1 promotes tumorigenesis by blocking the uL18 (universal large ribosomal subunit protein 18)-MDM2-p53 pathway in human cancer. *eLife* 7:e31275.
- Lim DS, Kim ST, Xu B, Maser RS, Lin J, Petrini JH, Kastan MB (2000) ATM phosphorylates p95/nbs1 in an S-phase checkpoint pathway. *Nature* 404(6778): 613-617.

- Lv BB, Ma RR, Chen X, Zhang GH, Song L, Wang SX, Wang YW, Liu HT, Gao P (2020) E2F1-activated SPIN1 promotes tumor growth via a MDM2-p21-E2F1 feedback loop in gastric cancer. *Mol Oncol* 14(10): 2629-2645.
- Matsuoka (1998) Linkage of ATM to Cell Cycle Regulation by the Chk2 Protein Kinase. *Science* 282(5395): 1893-1897.
- Yang F, Chen J, Liu B, Gao G, Sebastian M, Jeter C, Shen J, Person MD, Bedford MT (2021) SPINDOC binds PARP1 to facilitate PARylation. *Nature communications* 12(1): 6362.
- Yarden RI, Pardo-Reoyo S, Sgagias M, Cowan KH, Brody LC (2002) BRCA1 regulates the G2/M checkpoint by activating Chk1 kinase upon DNA damage. *Nature Genetics* 30(3): 285-289.
- Yuan H, Zhang P, Qin L, Chen L, Shi S, Lu Y, Yan F, Bai C, Nan X, Liu D, Li Y, Yue W, Pei X (2008) Overexpression of SPINDLIN1 induces cellular senescence, multinucleation and apoptosis. *Gene* 410(1): 67-74.
- Zhang P, Cong B, Yuan H, Chen L, Lv Y, Bai C, Nan X, Shi S, Yue W, Pei X (2008) Overexpression of spindlin1 induces metaphase arrest and chromosomal instability. *Journal of cellular physiology* 217(2): 400-408.
- Zhao F, Deng Y, Yang F, Yan Y, Feng F, Peng B, Gao J, Bedford MT, Li H (2024) Molecular Basis for SPINDOC-Spindlin1 Engagement and Its Role in Transcriptional Attenuation. *Journal of Molecular Biology* 436(7): 168371.
- Zhao Q, Qin L, Jiang F, Wu B, Yue W, Xu F, Rong Z, Yuan H, Xie X, Gao Y, Bai C, Bartlam M, Pei X, Rao Z (2007) Structure of human spindlin1. Tandem tudor-like domains for cell cycle regulation. *The Journal of biological chemistry* 282(1): 647-656.

Referee#2:

In this study, the authors have demonstrated involvement of SPIN1 in HR-mediated double strand break repair and show evidence supporting SPIN1 promotes association of Tip60 to H3K9me3 generated after DNA damage. They propose that SPIN1 association with H3K9me3 protects this mark against demethylase activity and serves to promote downstream activation of ATM and HR DNA damage repair. They identify SPIN1 as a potential target to increase sensitivity of cells to DNA damaging therapeutic agents such as ionizing radiation and chemotherapy. Overall the data shown generally supports their conclusions with modest effects of chemo- and radio-sensitization in the tested model.

Response: Thank you for the positive comments! In order to strengthen our conclusions, we have conducted additional mouse xenograft experiments during the revision process to investigate the effects of chemo-sensitization *in vivo*. We have included the corresponding figures (Fig. 5E-J) to illustrate these findings. The specific questions have been addressed below.

Revisions suggested:

-Figure 1 should include labels to clearly delineate which constructs represent full-length vs. specific tudor domain deletions

Response: Thank you for your suggestion. We sincerely apologize for the confusion caused by the lack of clarity in Figure 1. To address this issue, we have redrawn the schematic diagram in Figure 1C, ensuring that clear labels are included to differentiate between the full-length construct and specific tudor-like domain deletion mutants.

Figure for referees not shown.

-The authors suggest that SPIN1-H3K9me3-Tip60 form a complex promoting ATM activation and DNA damage repair and that SPIN1 is recruited by PAR binding, however it is unclear what form of PAR they are studying. Poly ADP ribosylation exists on multiple substrate types including PARP1 itself and histones. The paper requires better characterization of the PAR that was utilized in experiments shown in Figure 2. The methods section lists this reagent as a gift without analytics;

characterization should be provided in the supplement to allow for better understanding of and reproduction of the experiment.

Response: Thank you for your suggestion. We sincerely apologize for any confusion regarding poly(ADP-ribose) (PAR) used in dot blotting assay presented in Figure 2. In our study, we utilized the protein-free PAR, which was synthesized on a large scale using an enzymatic method. This protein-free PAR produced has been widely used as a standard, substrate, or binding partner in a variety of ADPR-related assays in relevant studies(Barkauskaite *et al*, 2013; Kam *et al*, 2018; Lin *et al*, 2018; Tan *et al*, 2012). We apologized for the omission of description about the biosynthesis and purification of PAR in our original manuscript. In the revised version, we have included a detailed description in Methods and Materials section (Page 19, Line 11-21). Briefly, a 20 ml incubation buffer containing 100 mM Tris-HCl pH 7.8, 10 mM MgCl₂, 1 mM NAD⁺, 1 mM DTT, 60 µg calf thymus histone, 50 µg octameric ‘activator’ oligonucleotide GGAATTCC, and 1.2 mg PARP1. The mixture was incubated at 30 °C for 60 min and stopped by adding 20 ml of ice-cold 20% TCA. DNase I was then used to remove Oligo DNA and proteinase K was employed to digest proteins. After extraction with phenol-chloroform-isoamyl alcohol, PAR was precipitated with ethanol overnight. The PAR polymer was then dissolved and finally purified by anion exchange chromatography.

-Figure 2: pulldown assays should include input lanes

Response: We appreciate your comment and apologize for the oversight in the initial submission. As suggested, we have conducted additional pulldown assays, and have included input lanes in Figure2(Fig. 2G).

Figure for referees not shown.

-Evidence for direct binding between SPIN1 and PAR and discussion of binding kinetics would be strengthened with orthogonal binding experiment data e.g. SPR, ITC; alternatively, would suggest that body and figure text should be rephrased to

reflect co-IP experiments e.g. refer to increased pulldown efficiency instead of increased binding affinity as kD values are not reported.

Response: Thank you for your suggestion. In our study, we performed *in vitro* pull-down experiments and *in vivo* co-IP assays to examine the binding of SPIN1 with PAR, not using binding experiments such as SPR or ITC to provide more robust data on the binding kinetics. Given the limited revision period, in our revised manuscript we have rephrased the body and figure text to reflect the pulldown and co-IP experiments. Specifically, we have described “indicating the enhanced pulldown efficiency associated with the presence of IDR”(Page7, Line 3-4). We have included a discussion section, highlighting the need for further SPR and ITC experiments to strengthen evidence of direct binding between SPIN1 and PAR (Page13, Line 1-4).

-Figure 3 should quantify the knockdowns as shown on WB in addition to flow cytometry

Response: Thank you for your suggestion. We have made revisions to Figure 3A and B by including the quantification of the knockdowns as shown on the Western blot analysis.

Figure for referees not shown.

-Regarding Figure 3: text or legend should define the difference between SPIN1-1 and SPIN1-2 (the methods clarify that these two different siRNA sequences)

Response: Thank you for your suggestion, and we apologize for any confusion caused. To address this issue, we have included a detailed description in the Materials and Methods section (Page 16, Line 8-10) that clearly states the target regions of the siRNA sequences. Specifically, siSPIN1-1 is designed to target the 5'UTR of the SPIN1 mRNA, while siSPIN1-2 is designed to target the CDS region of the SPIN1 mRNA.

-Figures implying kinetics should fit data to a curve or plot quantification data separately e.g. Figure 4D, Figure 5B-D

Response: Thank you for your comment, and we sincerely apologize for the errors in the quantification analysis. To address this issue, we have conducted a thorough re-

analysis of the quantification data, ensuring its accurate representation. We have made revisions to Figure 4D to reflect the correct kinetics of the data. In response to the suggestion made by Reviewer#1, we have performed additional experiments to include the siSPIN1 group as a control in the survival assay. Consequently, we have quantified the updated data in Figure 5B.

Figure for referees not shown.

Figure for referees not shown.

-Further discussion of the role of related protein SPINDOC is lacking; although this related protein is briefly mentioned in the text, available structural data and previous work outlining the role of SPINDOC in DNA damage response should be cited, if not also included as part of the model in the final figure.

citations below:

<https://doi.org/10.1016/j.jmb.2023.168371>

Response: Thank you for your valuable suggestion regarding the papers on SPINDOC. As suggested, we have incorporated the citations in our revised version. SPINDOC, also known as C11orf84, is closely associated with SPIN1. Previous studies have reported two distinct roles of SPINDOC in regulating SPIN1. Firstly, it regulates the transcriptional activator activity of SPIN1 by inhibiting its histone methyl-binding ability, thereby repressing the expression of SPIN1-regulated genes and the SPIN1-mediated activation of the Wnt signaling pathway (Bae *et al*, 2017a). Secondly, as recommended by the reviewer, a study (PMID: 34737271) highlights an alternative role of SPINDOC that is independent of SPIN1. This study demonstrates that SPINDOC interacts with PARP1 and facilitates its ADP-ribosyltransferase activity in response to DNA damage, indicating that SPINDOC has a SPIN1-independent role in regulating PARP1-mediated PARylation and the DNA damage response (Yang *et al*, 2021).

In 2021, Du *et al.* determined the crystal structure of the SPIN1 (aa:50-262)-C11orf84 (aa:253-295)-H3K4me3K9me3 (A1-G12) ternary complex (PDB code: 7CNA). They revealed that the C11orf84 fragment binds to the Tudor 3 domain of SPIN1, while the methyl histone peptide spans across the Tudor 1 and 2 domains. Furthermore, Du *et al.* showed that C11orf84 binding stabilizes SPIN1 and enhances its association with the bivalent H3K4me3K9me3 mark, thereby facilitating SPIN1's activation in rRNA transcription (Du *et al*, 2021). Recently, Zhao *et al.* resolved the crystal structures of SPIN1 (aa:50-262) bound to SPINDOCpep3 (aa:256-281) (PDB code: 7E9M) and SPINDOCpep2 (aa:228-239) (PDB code: 7EA1), respectively (Zhao *et al*, 2024). The SPIN1-SPINDOCpep3 complex revealed that SPINDOCpep3 (aa:256-281) interacts with the Tudor-like 3 domain of SPIN1, consistent with the structural analysis reported by Du *et al.* Furthermore, they found that two neighboring K/R-rich motifs (SPINDOCpep1 and SPINDOCpep2) bind to the acidic surface of SPIN1's Tudor-like 2 domain, thereby attenuating SPIN1's binding to the methylated H3 and its transcriptional co-activator activity (Zhao *et al*, 2024). As a result, these studies present contradictory results regarding the regulatory role of SPINDOC on SPIN1, particularly in enhancing SPIN1's association with methylated H3.

In our revised manuscript, we have included the citations of the available structural data and previous studies that highlight the involvement of SPINDOC in the DNA damage response. Additionally, considering that SPINDOC binds PARP1 to facilitate PARylation and DNA damage response independent of SPIN1 (Yang *et al*, 2021), we have included a discussion on the potential implications of SPINDOC binding to SPIN1 in the context of SPIN1's involvement in DNA damage repair (Page 14, Line 20-29). However, further research is needed to explore the precise regulatory mechanism of SPINDOC in SPIN1's involvement in DNA damage repair. Due to the contradictory findings regarding SPINDOC's role in SPIN1, we did not include SPINDOC in the model in the final figure.

-The authors do acknowledge that small molecule inhibitors of SPIN1 have been published and discuss future experiments with them, but if these are available to them presently, they would provide additional support. A suggested experiment would be treatment .with SPIN1 inhibitors in a time course following DNA damaging treatment to understand the kinetics of of SPIN1 mediated DNA damage repair.

Response: Thank you for your suggestion. Currently, there are several small molecule inhibitors of SPIN1 that have been reported, most of which were synthesized by the authors themselves. Notably, EML405, EML631, VinSpin1N, and A366 have been shown interactions with Tudor-like 1 and Tudor-like 2 domains, while MS31 specifically binds to Tudor-like 1 domain (Bae *et al*, 2017b; Fagan *et al*, 2019; Li *et al*, 2021). The Tudor-like 2 domain of SPIN1 recognizes H3K4me3 or H4K20me3, while the Tudor-like 1 can recognize H3R8me2a or H3K9me3. In our study, as the interaction between SPIN1 and H3K9me3 is crucial for SPIN1-mediated HR repair, we require an inhibitor that can effectively disrupt the interaction between SPIN1 Tudor-like 1 domain and H3K9me3. Initially, we attempted to obtain some of these inhibitors from <https://www.thesgc.org/chemical-probes>, as recommended in a published paper by Fagan, V. *et al.*(Fagan *et al*, 2019). However, we were informed that these inhibitors were not currently available. Additionally, we contacted the authors who reported several SPIN1 inhibitors targeting both Tudor-like 1 and 2 domains, but unfortunately, we did not receive a response from them. Finally, we contacted the MCE company for SPIN1 inhibitors and found that two inhibitors, MS31 trihydrochloride and A366, were available. However, MS31 trihydrochloride disrupts the interaction between SPIN1 and H3K4me3 by targeting the Tudor-like 2 domain of SPIN1. Therefore, it is not suitable for our specific research objectives. Consequently, we have purchased A366 for our additional experiments.

A366 is known as a dual inhibitor of SPIN1 and G9a/GLP, exhibiting potent but nonselective inhibition of SPIN1. Two A366 molecules bind to one SPIN1 molecule, occupying both Tudor-like 1 and 2 domains. Wagner *et al.* initially identified A366 as a small-molecule ligand of a Tudor domain containing methyl lysine reader. A366 was tested in HL-60 cells at a concentration of 100 μ M for cell penetration and target engagement assays (Wagner *et al*, 2016). However, we observed poor cell growth and abnormal cell morphology when U2OS cells were treated with this concentration. Thus, we attempted three different concentrations of A366 (2 μ M, 20 μ M, 100 μ M) to treat U2OS cells and conducted a micro-irradiation assay to measure the kinetics of SPIN1-mediated DNA damage repair. Surprisingly, our result showed that GFP-SPIN1 was still recruited to DNA damage sites after A366 treatment (Fig. R1). Based on these findings, it appears that A366 does not affect SPIN1's recruitment to DNA damage sites. However, due to A366 being a nonselective SPIN1 inhibitor that also targets G9a/GLP and binds both Tudor-like 1 and 2 domains, we are unable to conclusively determine whether SPIN1 inhibitors prevent SPIN1 from being recruited to DNA damage sites or not. Further research is required to investigate the potential applications of SPIN1 small molecule inhibitors that specifically target the Tudor-like

1 domain of SPIN1 and their roles in SPIN1's involvement in DNA damage repair.

Figure for referees not shown.

Figure R1. Kinetics of GFP-SPIN1 relocation to DNA damage sites after A366 treatment. (A) U2OS cells expressing GFP-tagged SPIN1 were treated with or without A366 and the cell morphology was observed using microscopy. Scale bar =20 μ m. (B) The relocation kinetics were monitored in a time course. Kinetic analysis was performed using CellSens software (Olympus). The data represents the mean \pm SD. Scale bar =5 μ m.

Reference:

- Bae N, Gao M, Li X, Premkumar T, Sbardella G, Chen J, Bedford MT (2017a) A transcriptional coregulator, SPIN.DOC, attenuates the coactivator activity of Spindlin1. *The Journal of biological chemistry* 292(51): 20808-20817.
- Bae N, Viviano M, Su X, Lv J, Cheng D, Sagum C, Castellano S, Bai X, Johnson C, Khalil MI, Shen J, Chen K, Li H, Sbardella G, Bedford MT (2017b) Developing Spindlin1 small-molecule inhibitors by using protein microarrays. *Nature chemical biology* 13(7): 750-756.
- Barkauskaite E, Brassington A, Tan ES, Warwicker J, Dunstan MS, Banos B, Lafite P, Ahel M, Mitchison TJ, Ahel I (2013) Visualization of poly(ADP-ribose) bound to PARG reveals inherent balance between exo- and endo-glycohydrolase activities. *Nature communications* 4: 2164
- Du Y, Yan Y, Xie S, Huang H, Wang X, Ng RK, Zhou M-M, Qian C (2021) Structural mechanism of bivalent histone H3K4me3K9me3 recognition by the Spindlin1/C11orf84 complex in rRNA transcription activation. *Nature communications* 12(1):949.
- Fagan V, Johansson C, Gileadi C, Monteiro O, Dunford JE, Nibhani R, Philpott M, Malzahn J, Wells G, Faram R, Cribbs AP, Halidi N, Li F, Chau I, Greschik H, Velupillai S, Allali-Hassani A, Bennett J, Christott T, Giroud C, Lewis AM, Huber KVM, Athanasou N, Bountra C, Jung M, Schule R, Vedadi M, Arrowsmith C, Xiong Y, Jin J, Fedorov O, Farnie G, Brennan PE, Oppermann U (2019) A Chemical Probe for Tudor Domain Protein Spindlin1 to Investigate Chromatin Function. *Journal of medicinal chemistry* 62(20): 9008-9025.

Referee #3:

SPIN1 is known to be a reader of H3K9me3. Histone H3K9 methylation links DNA damage detection to activation of Tip60. Based on these background, in this study, authors revealed a new function of SPIN1 in binding to DNA damage sites. Firstly, they reported that SPIN1 is recruited to DNA lesions through its N-terminal disordered region that binds to PAR, and facilitates homologous recombination (HR)-mediated DNA damage repair. Secondly, SPIN1 maintains the stability of H3K9me3 and enhances the interaction between H3K9me3 and Tip60, leading to the activation of ATM and HR repair. Finally, authors demonstrate that SPIN1 contributes to chemoresistance by enhancing DNA repair. However, the detail mechanism underlying the epigenetic changes is totally unknown. In addition, some issues such as lack of appropriate controls or approaches preclude publication for EMBO reports.

Response: Thank you for the comments! The specific questions have been addressed below.

Comments:

1. Authors state that SPIN1 preserves the presence of H3K9me3 at the site of damage, however, the exact mechanism is absent, although lysine demethylase KDM4B is mentioned. How does SPIN1 regulate KDM4B? Is KDM4B involved in the regulation of SPIN1 on H3K9me3?

Response: Thank you for your comment. We apologized for the limited information provided regarding the exact mechanism by which SPIN1 regulates KDM4B and the involvement of KDM4B in the regulation of SPIN1 on H3K9me3. In our working model, we speculate that SPIN1 binding to H3K9me3 temporarily prevents KDM4B from demethylating H3K9me3. This, in turn, facilitates the interaction between H3K9me3 and Tip60, finally leading to the activation of ATM and HR repair. We have indicated the possible role of KDM4B with a dotted line in our working model to denote that it is purely hypothetical. We apologized for any confusion caused.

To address this concern, we have performed additional experiments to investigate whether SPIN1 regulates KDM4B. We examined the expression of KDM4B protein by Western blot assay upon overexpressing SPIN1. The results showed that, with or without IR treatment, SPIN1 had no discernible effect on the expression of KDM4B protein compared to the control(Fig. R2). This indicates that the maintenance of H3K9me3 stability by SPIN1 following DNA damage is not likely due to the regulation of KDM4B protein levels. Therefore, our speculation regarding the potential regulation of H3K9me3 by SPIN1 through KDM4B requires more investigation. We have included a question mark in this portion of the final working model in our revised manuscript.

Figure for referees not shown.

Figure R2. The impact of SPIN1 on KDM4B protein levels following DNA damage. HEK293T cells expressing either SPIN1 or a control vector were treated with or without IR. Western blot analysis was performed with the indicated antibodies.

2. *Authors confirm that SPIN1 binds to PAR dependent of the N-terminus IDR. Authors also mentioned in the introduction that the first Tudor module of SPIN1 can recognize H3R8me2a or H3K9me3. Does SPIN1 bind to PAR or H3K9me3 through different domain? Does PAR inhibitor influence the interaction of SPIN1 and H3K9me3?*

Response: Thank you for your valuable comment. Based on our findings from micro-irradiation assay and pull-down experiments, we have identified that the IDR region of SPIN1 plays a crucial role in binding to PAR. Previous studies and available structural data have clearly reported that the Tudor-like 1 domain of SPIN1 interacts with H3K9me3 (Du *et al*, 2021; Zhao *et al*, 2020). Therefore, SPIN1 binds to PAR and H3K9me3 through different domains or regions within the protein. Regarding the influence of the PAR inhibitor Olaparib on the interaction between SPIN1 and H3K9me3, our results showed that Olaparib treatment abolished the recruitment of SPIN1 to DNA damage sites (Fig. 2B). This suggests that Olaparib can affect the interaction between SPIN1 and H3K9me3 at DNA damage sites.

We performed additional Co-IP assays to explore this interaction in the context of Olaparib treatment. However, the results showed that Olaparib treatment did not significantly affect the overall interaction between SPIN1 and H3K9me3 in cells (Fig. R3). This discrepancy may be because the Co-IP assay examines the overall relationship between the target proteins. Considering that Olaparib influences the recruitment of SPIN1 to DNA damage sites (Fig. 2B), it is likely that Olaparib affects the interaction between SPIN1 and H3K9me3 at DNA damage sites. However, SPIN1 and H3K9me3 proteins that are not localized at DNA damage sites may still interact with each other since SPIN1 binds to H3K9me3 through its Tudor-like 1 domain. Further investigation is required to fully understand the impact of Olaparib on the interaction of SPIN1 and H3K9me3.

Figure for referees not shown.

Figure R3. The influence of PAR inhibitor on the interaction between SFB-SPIN1 and H3K9me3 via Co-IP assay. HEK293T cells expressing either SPIN1 or a control vector were treated with or without Olaparib, and then subjected to co-immunoprecipitation using streptavidin beads. Western blot analysis was performed with the indicated antibodies.

3. Authors stated that SPIN1 enhances chemoresistance by cell viability assay. Evidence in vivo is necessary for this conclusion. For example, both cisplatin and olaparib are used to treat the mice implanted of control or SPIN1-knockdown cells, and tumor sizes will be measured.

Response: Thank you for your suggestion. We agree that providing *in vivo* evidence would strengthen our conclusion regarding the role of SPIN1 in enhancing chemoresistance. To address this concern, we performed *in vivo* experiments using a mouse model. First, we generated stable cell lines in SGC7901 cells with either SPIN1 knockdown (shSPIN1) or control (shNC) and confirmed the efficiency of SPIN1 knockdown (Fig. EV3C). Then we implanted these control cells and SPIN1-knockdown cells into male NTG mice. We inject Cisplatin (3 mg/kg/two days) or Olaparib (50 mg/kg/day) for 14 days after the tumor forms for 10 days. To assess the impact of SPIN1 on chemoresistance, we monitored tumor sizes and weights throughout the experiment. Our results showed that SPIN1 contributed to cancer chemoresistance *in vivo* (Fig. 5E-J and Fig. EV4D, E). This finding suggests that the enhanced efficiency of DNA DSB repair may be responsible for the observed chemoresistance.

4. A one-way ANOVA with a non-parametric post hoc test or two-way ANOVA should be used to analyze the data rather than multiple t-tests. For example, Fig2B, Fig4D, Fig5. The detail of statistical method should be included in each figure legend.

Response: Thank you for your valuable suggestion. To address this concern, we have reanalyzed the data using appropriate statistical methods as suggested. We have performed one-way ANOVA followed by the Tukey Kramer test for the data presented in Figure 2B, 4D and Figure 5A-D,F,G,I,J. And we have included the statistical analysis details in the relevant figure legends.

Further comments from referee 3:

In this study, the authors reveal a new function of SPIN1 in binding to DNA damage sites, although the precise mechanism is not well described (comments1 and 2). Here, the important function of SPIN1 in DNA repair is to participate in chemoresistance. However, the related conclusion is overstated, and further experiments are required (comments3). In addition, there are some issues with the appropriateness of the statistical analysis that should be addressed (comments4). Overall, as the standards of EMBOR, if the authors could reasonably respond to the comments by additional experiments or discussion, this manuscript can be in major revision.

Response: Thank you for your valuable comments. We have made the following revisions to our manuscript during the revision process:

1) We have revised the working model and provided a more detailed explanation of the proposed mechanism, along with the evidence that is currently available. This includes the addition of more experiments and citations that study the role of SPIN1's binding partner, SPINDOC, in response to DNA damage, as suggested by three Reviewers in the Discussion section. We found that following DNA damage, the protein levels of KDM4B are unaffected by SPIN1 overexpression. Therefore, we have marked it as a hypothetical possibility with a question mark in the working model. Meanwhile, we have emphasized the need for additional research to elucidate the precise molecular mechanisms involved and also acknowledged the limitations that require further investigation (Page14,Line 22-29).

2) We have strengthened the role of SPIN1 in chemoresistance using *in vivo* animal experiments. These experiments provide *in vivo* evidence that supports the role of SPIN1 in chemoresistance (Page11,Line 20-29).

3) To address the concerns regarding the appropriateness of the statistical analysis, we have reanalyzed the data using appropriate statistical methods, specially one-way ANOVA followed by the Tukey Kramer test. We provided the statistical analysis details in the relevant figure legends.

We appreciate your constructive and valuable comments. Hopefully, these responses we included in the revised version of our manuscript can strengthen the conclusions of our manuscript, thereby meeting the EMBOR requirements.

Reference:

Du Y, Yan Y, Xie S, Huang H, Wang X, Ng RK, Zhou M-M, Qian C (2021) Structural mechanism of bivalent histone H3K4me3K9me3 recognition by the Spindlin1/C11orf84 complex in rRNA transcription activation. *Nature communications* 12(1):949.

Zhao F, Liu Y, Su X, Lee JE, Song Y, Wang D, Ge K, Gao J, Zhang MQ, Li H (2020) Molecular basis for histone H3 "K4me3-K9me3/2" methylation pattern readout by Spindlin1. *Journal of Biological Chemistry* 295(49):16877-16887.

Dear Prof. Wu

Thank you for the submission of your revised manuscript. We have now received the enclosed reports from all referees, and I am happy to say that all support its publication now. Only a few editorial requests will need to be addressed before we can proceed with the official acceptance of your manuscript:

- Please reduce the number of keywords to 5.
- Please correct the conflict of interest subheading to "Disclosure and Competing Interests Statement"
- The FUNDING INFO needs to be part of the Acknowledgments. The following info needs to be added when the new ms version is uploaded into our online system: S&T Program of Hebei (No.216Z2602G), Interdisciplinary Research Program of Natural Science of Hebei University (DXK202006).
- Figure 5 is split in 2 but all figures need to fit on one single page. Please correct.
- The names of the source files for EV figures need to be updated: Figure EV1 instead of Supplementary 1, etc.
- All source data need to be uploaded as one (zipped) file per main figure. All source data for Expanded View figures need to be zipped together into one single file.
- Materials and Methods need to be renamed to just Methods.
- Please define the annotated p values ***/**/* as well as provide the exact p-values for the same in the legend of figure 2b; 3a-b, f-g, j, l, n; 4a, c-d, h; 5a-b, c-d, f-g, i-j; EV 2a; EV 3b; as appropriate.
- Please indicate the statistical test used for data analysis in the legend of figure EV 3b.
- Please note that the box plots need to be defined in terms of minima, maxima, centre, bounds of box and whiskers, and percentile in the legend of figure EV 4b.
- Please note that the error bars are not defined in the legends of figures 4a, c-d, h; EV 3b.
- Please note that the white arrowheads are not defined in the legend of figure 2b. This needs to be rectified.

EMBO press papers are accompanied online by A) a short (1-2 sentences) summary of the findings and their significance, B) 2-3 bullet points highlighting key results and C) a synopsis image that is exactly 550 pixels wide and 200-600 pixels high (the height is variable). The synopsis image should provide a sketch of the major findings, like a graphical abstract. Please note that text needs to be readable at the final size. Please send us this information along with the final manuscript.

I would like to suggest some minor changes to the title and abstract. Please let me know whether you agree with the following:

SPIN1 enhances chemoresistance and Tip60 binding to H3K9me3 to facilitate HR repair

The tandem Tudor-like domain-containing protein Spindlin1 (SPIN1) is a transcriptional coactivator with critical functions in embryonic development and emerging roles in cancer. However, the involvement of SPIN1 in the DNA damage repair has remained unclear. Our study shows that SPIN1 is recruited to DNA lesions through its N-terminal disordered region that binds to Poly-ADP-ribose (PAR), and facilitates homologous recombination (HR)-mediated DNA damage repair. SPIN1 promotes H3K9me3 accumulation at DNA damage sites and enhances the interaction between H3K9me3 and Tip60, thereby promoting the activation of ATM and HR repair. We also show that SPIN1 increases chemoresistance. These findings reveal a novel role for SPIN1 in the activation of H3K9me3-dependent DNA repair pathways, and suggest that SPIN1 may contribute to cancer chemoresistance by modulating the efficiency of double-strand break (DSB) repair.

Kind regards,
Esther

Referee #1:

The authors have addressed all the issues raised and the manuscript is much improved. There are no outstanding issues.

Referee #2:

The authors have revised the manuscript in response to the initial reviews, and it is now improved.

Referee #3:

The authors well addressed my concerns.

All editorial and formatting issues were resolved by the authors.

Prof. Chen Wu
Hebei University
Wusi East Road
Baoding, Hebei Province 071002
China

Dear Prof. Wu,

I am very pleased to accept your manuscript for publication in the next available issue of EMBO reports. Thank you for your contribution to our journal.

Yours sincerely,
